# Actin is an evolutionarily-conserved damage-associated molecular pattern that signals tissue injury in *Drosophila melanogaster*

Naren Srinivasan[1†], Oliver Gordon[1†], Susan Ahrens[1‡], Anna Franz[2], Safia Deddouche[1§], Probir Chakravarty[3], David Phillips[4], Ali A Yunus[5], Michael K Rosen[5], Rita S Valente[6], Luis Teixeira[6], Barry Thompson[7], Marc S Dionne[8], Will Wood[9], Caetano Reis e Sousa[1*]

[1]Immunobiology Laboratory, The Francis Crick Institute, London, United Kingdom; [2]Department of Biochemistry, Biomedical Sciences, University Walk, University of Bristol, Bristol, United Kingdom; [3]Bioinformatics, The Francis Crick Institute, London, United Kingdom; [4]Genomics-Equipment Park, The Francis Crick Institute, London, United Kingdom; [5]Department of Biophysics, University of Texas Southwestern Medical Center, Dallas, United States; [6]Instituto Gulbenkian de Ciência, Oeiras, Portugal; [7]Epithelial Biology Laboratory, The Francis Crick Institute, London, United Kingdom; [8]Department of Life Sciences and MRC Centre for Molecular Bacteriology and Infection, South Kensington Campus, Imperial College London, London, United Kingdom; [9]Department of Cellular and Molecular Medicine, Biomedical Sciences, University of Bristol, Bristol, United Kingdom

*For correspondence: caetano@crick.ac.uk

[†]These authors contributed equally to this work

Present address: [‡]Voisin Consulting Life Sciences, Surrey, United Kingdom; [§]Open Innovation Access Platform, Sanofi Strasbourg, Strasbourg, France

Competing interests: The authors declare that no competing interests exist.

**Abstract** Damage-associated molecular patterns (DAMPs) are molecules released by dead cells that trigger sterile inflammation and, in vertebrates, adaptive immunity. Actin is a DAMP detected in mammals by the receptor, DNGR-1, expressed by dendritic cells (DCs). DNGR-1 is phosphorylated by Src-family kinases and recruits the tyrosine kinase Syk to promote DC cross-presentation of dead cell-associated antigens. Here we report that actin is also a DAMP in invertebrates that lack DCs and adaptive immunity. Administration of actin to *Drosophila melanogaster* triggers a response characterised by selective induction of STAT target genes in the fat body through the cytokine Upd3 and its JAK/STAT-coupled receptor, Domeless. Notably, this response requires signalling via Shark, the *Drosophila* orthologue of Syk, and Src42A, a *Drosophila* Src-family kinase, and is dependent on Nox activity. Thus, extracellular actin detection via a Src-family kinase-dependent cascade is an ancient means of detecting cell injury that precedes the evolution of adaptive immunity.

## Introduction

Trauma, burns, ischemia, strenuous exercise, all induce a sterile inflammatory response. It is likely that this response evolved to clear cell debris, promote tissue repair and maintain tissue sterility (*Zelenay and Reis e Sousa, 2013*; *Eming et al., 2014*) but, if uncontrolled, it can lead to (aseptic) shock and, in some cases, death (*Rock et al., 2010*). The prevailing notion is that sterile inflammation is initiated by pro-inflammatory signals that are released by damaged cells. These include intracellular components that are exposed when cells lose their membrane integrity, such as ATP, uric

**eLife digest** All animals must be able to detect and repair injuries quickly. To do this, the body triggers a process called inflammation at the site of injury to remove dead and damaged cells, keep the area free from infection and trigger repair. However, if an area becomes excessively inflamed, or remains inflamed for a long period of time, it can contribute to diseases like cancer and Alzheimer's disease.

Inflammation starts when the body detects molecules that are released when cells die or are damaged to the extent that they become leaky. Actin, which is a protein that usually provides structural support to the cell, is one molecule that is sensed by the immune system in mammals when released from dying cells. However, it is not clear whether released actin can also trigger reactions in simpler animals like fruit flies.

To address this question, Srinivasan, Gordon et al. injected fruit flies with actin. These animals developed a widespread reaction reminiscent of inflammation seen in mammals. Further experiments showed that actin switches on a signalling pathway called JAK/STAT, which is known to become active when flies experience other types of stress. The JAK/STAT proteins activate a signalling pathway that leads to changes in gene activity. Srinivasan, Gordon et al. also showed that part of a cascade of signals triggered by released actin in mammals is shared in fruit flies.

These findings suggest that this important response to actin evolved a long time ago. A future challenge is to find out how the body detects and mops up released actin, which may help us to understand how actin can contribute to various inflammatory diseases.

acid, RNA and DNA, collectively known as damage-associated molecular patterns (DAMPs) (*Rock et al., 2010*; *Zelenay and Reis e Sousa, 2013*). The universe of DAMPs and their receptors, as well as the mechanisms regulating DAMP responses, remains underexplored. This is partly because early research in this area was tainted by issues of microbial contamination (*Beg, 2002*) and because immunologists have often focussed on sterile inflammation from the narrow perspective of adaptive immunity (*Matzinger, 1994*; *Land et al., 1994*). However, it is probable that responses to DAMPs, like responses to microbes, pre-date the vertebrate evolution of T and B cells and have an early metazoan origin, much like the clearance of dead cells (*Krysko et al., 2011*; *Hochreiter-Hufford and Ravichandran, 2013*; *Eming et al., 2014*). Therefore, the study of invertebrate responses to DAMPs could offer a different perspective into the induction of sterile inflammation, akin to how research into insect immunity to infection led to the identification of Toll signalling and paved the way to the discovery of an analogous pathway in vertebrates (*Lemaitre et al., 1996*).

The immune system of *Drosophila melanogaster* has been widely studied in the context of infection. It consists of a cellular and a humoural arm, in addition to cell-intrinsic antiviral RNAi responses (*Buchon et al., 2014*; *Lemaitre and Hoffmann, 2007*). The cellular arm is made up of three macrophage-like types of cells, collectively termed haemocytes (*Buchon et al., 2014*; *Lemaitre and Hoffmann, 2007*). The humoural immune response relies on antimicrobial peptides (AMPs) that are synthesised in the fat body (the fly equivalent of the liver) and then secreted into the haemolymph to provide systemic protection from bacteria and fungi (*Lemaitre and Hoffmann, 2007*; *Buchon et al., 2014*). The production of AMPs is regulated by two different pathways. The Toll pathway is activated by peptidoglycan fragments of Gram-positive bacteria, fungal β-glucans, and pathogen-derived protease activity in the haemolymph (*Buchon et al., 2014*; *Lemaitre and Hoffmann, 2007*). The Imd pathway is activated by peptidoglycan fragments from Gram-negative bacteria (*Buchon et al., 2014*; *Lemaitre and Hoffmann 2007*). Activation of either pathway results in the translocation of distinct NF-κB family transcription factors into the nucleus and the subsequent synthesis of AMPs best suited to neutralise the type of microorganism detected (*Lemaitre and Hoffmann, 2007*; *Buchon et al., 2014*). A third pathway contributing to *Drosophila* humoural immunity involves Janus Kinase/Signal Transducer and Activator of Transcription (JAK/STAT) signalling. In contrast to the Toll and Imd pathway, the JAK/STAT pathway has not yet been shown to be directly induced by sensors of invading microorganisms (*Myllymäki and Rämet, 2014*). However, it has been implicated in resistance to as well as tolerance to viral infections (*Dostert et al., 2005*;

Lamiable et al., 2016). Notably, the JAK/STAT pathway is activated by different types of stresses (e.g. heat, mechanical pressure, oxidative stress or UV irradiation) (Lemaitre et al., 1996; Ekengren et al., 2001; Ekengren and Hultmark, 2001). All of these insults likely result in cell death suggesting the possibility that JAK/STAT pathway activation might be triggered by DAMPs rather than microbes.

The JAK/STAT pathway is elicited by cytokines of the Unpaired (Upd) family – Upd1 (Harrison et al., 1998), Upd2 (Gilbert et al., 2005; Hombría et al., 2005) and Upd3 (Agaisse et al., 2003; Wright et al., 2011) – all of which serve as ligands for the only JAK/STAT-coupled receptor in Drosophila, Domeless (dome) (Brown et al., 2001). The binding of Upds induces Domeless dimerization and activation of a single JAK (termed Hopscotch). Activated Hopscotch proteins phosphorylate one another allowing for recruitment of the single Drosophila STAT family transcription factor, STAT92E. The latter is then phosphorylated by Hopscotch, resulting in dimerisation and translocation into the nucleus. STAT92E dimers bind to the promoters of their target genes (Kiu and Nicholson, 2012; Bina et al., 2010; Müller et al., 2005) including, amongst others, ones encoding proteins involved in viral resistance (Lamiable et al., 2016; Dostert et al., 2005), as well as proteins of the Turandot family such as Turandot M (TotM). The exact function of Turandot family proteins is not known but they have been controversially argued to be linked to stress resistance (Ekengren and Hultmark, 2001; Ekengren et al., 2001; Mahapatra and Rand, 2012; Zhong et al., 2013). Besides a role in host defence, the JAK/STAT pathway has also been linked to energy metabolism (Rajan and Perrimon, 2012) and regenerative processes, for example in the gut (Jiang et al., 2009). The involvement of JAK/STAT signalling in regeneration is particularly interesting given the role of DAMPs in contributing to tissue repair (Vénéreau et al., 2015).

We have previously identified DNGR-1 (also known as CLEC9A) as a vertebrate-restricted innate immune receptor dedicated to DAMP recognition (Sancho et al., 2009). DNGR-1 is phosphorylated by Src family kinases and then signals via Syk although it does not induce inflammation. Rather, DNGR-1 is expressed by dendritic cells (DCs) and signals to favour cross-presentation of antigens from dead cells, contributing to CD8[+] T cell responses to cytopathic infections and, possibly, tumours (Iborra et al., 2012; Zelenay et al., 2012; Sancho et al., 2009). We and others subsequently found that the DAMP recognised by DNGR-1 is F-actin, the polymer of G-actin that provides higher eukaryotic cells with structural integrity (Ahrens et al., 2012; Zhang et al., 2012). Actin is an ideal DAMP given that it is extremely conserved (90% identity between yeast and humans) and highly abundant and ubiquitous within all eukaryotic cells but absent from extracellular fluids. We therefore hypothesised that released actin constitutes an evolutionarily-conserved DAMP whose detection might involve a signalling pathway conserved from flies to mammals. This would be analogous to the conservation of the Toll signalling pathway (albeit not the upstream receptors) in the Drosophila and vertebrate response to fungi and bacteria. Here, we show systemic administration of actin to Drosophila selectively triggers a JAK/STAT response and that this requires the fly homologues of Src and Syk. Our data therefore reveal an evolutionarily-conserved tryosine kinase-based pathway for recognising damage through sensing of released or exposed actin.

## Results

### Injection of actin induces STAT target genes in Drosophila

To test whether actin might act as a DAMP in Drosophila, we injected $w^{1118}$ adult flies with actin or with buffer alone and carried out RNAseq gene expression profiling of total fly extracts (Figure 1a). Actin injection led to the differential expression of a large number of genes as compared to injection of buffer control: 241 genes were induced or repressed at 3 hr, 1297 genes at 6 hr and 351 genes at 24 hr post-injection (Figure 1b). Notably, among genes that were induced selectively in actin treated flies, we found the members of the Tot gene family including TotM, TotA and TotC, as well as Socs36E, Diedel and thioester-containing protein (Tep)1 (Figure 1b, c), all of which are STAT-dependent (Boutros et al., 2002; Müller et al., 2005; Lagueux et al., 2000). Gene set enrichment analysis (GSEA) of published datasets confirmed that target genes of STAT92E were highly enriched in actin- compared to buffer-injected flies (Figure 1d) (p<0.0001). The presence of STAT binding sites in promoters of genes upregulated by actin but not buffer injection was validated by bioinformatics

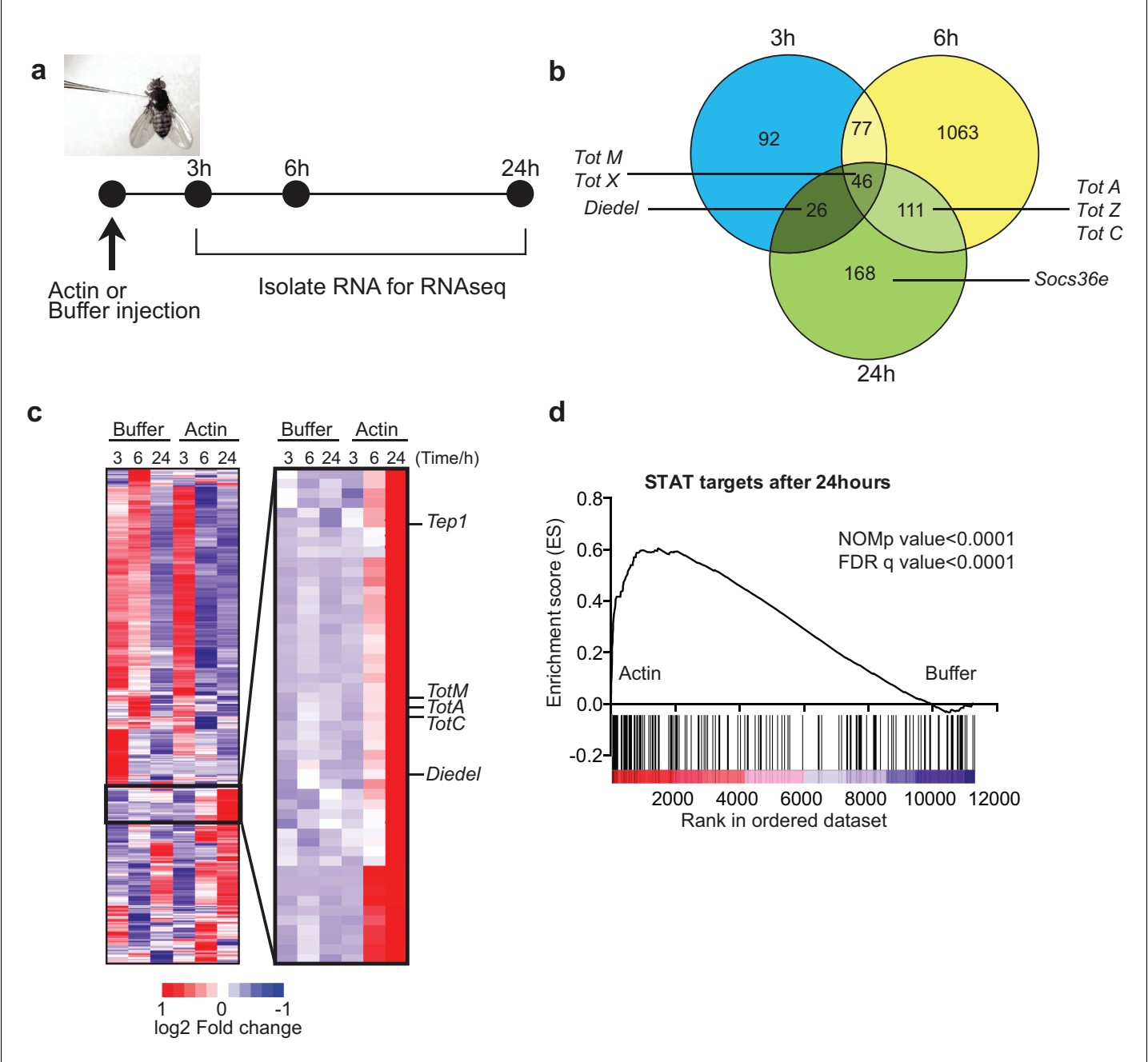

**Figure 1.** Global gene expression profiling reveals strong enrichment for JAK/STAT regulated genes upon actin injection. (a) Groups of control ($w^{1118}$) flies were injected with either buffer or actin before euthanasia at 3, 6 or 24 hr post injection. Flies within each group were pooled, RNA was isolated and processed for RNAseq analysis as described in the Methods. (b) Genes differentially expressed between buffer and actin-injected flies, including both up and down regulated transcripts, are represented for each time point in a Venn diagram. Only genes differentially expressed with a false discovery rate (FDR) < 0.05 were included in the analysis. The numbers within each set represent the numbers of genes differentially expressed. Select STAT target genes are indicated. (c) Differentially expressed genes between actin- and buffer-injected flies at 3, 6 and 24 hr were used to draw a hierarchical heat map. Genes were clustered using a Euclidean distance matrix and average linkage clustering. Samples were ordered based on time and treatment. The heat map shows the average expression values from triplicate samples. Red indicates higher expression and blue indicates lower expression relative to the mean expression of probes across all samples. The black box highlights genes for which there was the biggest fold change increase in actin- relative to buffer-injected flies. (d) Enrichment plot from GSEA showing that targets of STAT92E are enriched within the upregulated gene set in actin-injected files relative to buffer injected flies after 24 hr.

analyses (STAT responsive elements: p=$7.8 \times 10^{-4}$ using Transfac database, p=$1.63 \times 10^{-8}$ using JAS-PAR database).

We confirmed by RT-qPCR that actin but not buffer injection potently triggers expression of *Tot*, *Tep* and *Diedel* genes peaking at 24 hr after injection (*Figure 2a-f* and data not shown). Actin injection did not lead to the expression of the AMP genes *Drs* and *Dpt*, which were induced only upon experimental fungal or bacterial infection (*Figure 2g, h*). This indicates the absence of microbial contaminants in actin preparations and demonstrates that the response to actin differs from that to septic injury, in which induction of STAT targets is invariably accompanied by that of genes downstream of *Toll* or *Imd* (*Agaisse et al., 2003*; *Chakrabarti et al., 2016*; *Brun et al., 2006*). In this regard, *TotM* induction by actin was much greater than that elicited by injury (clean or septic) or by heat shock, which are classically regarded as inducers of the JAK-STAT signalling pathway (*Agaisse et al., 2003*; *Chakrabarti et al., 2016*) (*Figure 2—figure supplement 1a* and data not shown).

The uniqueness of the response to actin was further patent in the fact that injection of a plethora of sugars, salts, lipids, polysaccharides, proteins, peptides, or amino acids did not induce *TotM* (*Figure 2—figure supplement 2a–c*). Similarly, injection of ATP, DNA, uric acid (monosodium urate – MSU) or heat shock proteins (HSPs) – all of which act as DAMPs in mammals – failed to elicit *TotM* in flies (*Figure 2—figure supplement 2d*). The finding that *TotM* is not induced by any of the above stimuli other than actin needs to be tempered by the fact that they were all tested at a single, somewhat arbitrary, dose but it also acts as a control for the effects of injection-mediated injury. Importantly, the globular tertiary structure of actin was required for the response as injection of denatured protein failed to elicit *TotM* (*Figure 2—figure supplement 2e*). Finally, actin injection induced equivalent *TotM* levels in germ-free and SPF (specific pathogen free) flies, showing that microbiota or their products do not contribute to the response, despite the fact that they are inevitably introduced into the thorax of SPF flies upon piercing of the cuticle during injection (*Figure 2i*). Similarly, the response to actin did not require *Wolbachia*, a genus of vertically-transmitted intracellular bacteria that infect many arthropods and is known to impact immune responses (*Teixeira et al., 2008*; *Hedges et al., 2008*). Altogether, these data indicate that the introduction of actin into the haemolymph of flies uniquely induces a sterile inflammatory response characterised by selective induction of STAT target genes.

To ensure an experimental window for mechanistic dissection of the response (see below), most of the above experiments employed injection of 36.8 ng of actin, which is roughly equivalent to the amount contained in 3000 HeLa cells (data not shown). However, *TotM* induction could be observed upon injection of as little as 0.1 ng actin, which corresponds to the contents of 8–10 cells, thereby underscoring physiological relevance (*Figure 2j*). Actin from all tested species (human, rabbit and *Drosophila*) induced *TotM* (see Materials and Methods), consistent with the extreme evolutionary conservation of the protein. Importantly, *TotM* induction was not due to the cytopathic effects of actin filaments *in vivo* as the response was triggered equally by phalloidin-stabilised filamentous (F)- or latrunculin-stabilised globular (G)-actin (*Figure 2k*). The ability of G-actin to stimulate the response *in vivo* in the absence of polymerisation could be formally demonstrated by injecting a *Drosophila* non-polymerisable G-actin mutant. The monomeric nature of the G-actin mutant used was first verified by probing with mDNGR-1 extracellular domain, which specifically binds F- but not G-actin (*Figure 2—figure supplement 1b*) (*Ahrens et al., 2012*; *Hanč et al., 2015*). Injection of the G-actin mutant induced a response equivalent to that triggered by an injection of polymeric F-actin (*Figure 2l*).

## Extracellular actin-induced STAT activation occurs in the fat body

To further analyse the response, we focused on the tissues where STAT activation might occur. Analysis of physically-dissected body parts of $w^{1118}$ flies injected with actin revealed that STAT target gene expression was enriched in regions containing the fat body (*Figure 3a*). We confirmed that STAT activation takes place in the fat body by using reporter flies in which GFP is expressed under the control of a STAT response element and in which, additionally, fat body cells are labelled with Tomato fluorescent protein for ease of organ identification (*Stat92-dGFP+LPP*-Gal4>UAS-Myr-td-Tom). Injection of actin, but not mock injection (0 hr) or injection of buffer alone, resulted in a time-dependent increase in GFP fluorescence exclusively in Tomato⁺ fat body cells (*Figure 3b, c*). We next used a genetic approach to understand how extracellular actin leads to STAT activity in the fat body and to assess the contribution of that organ to the global response measured in total fly

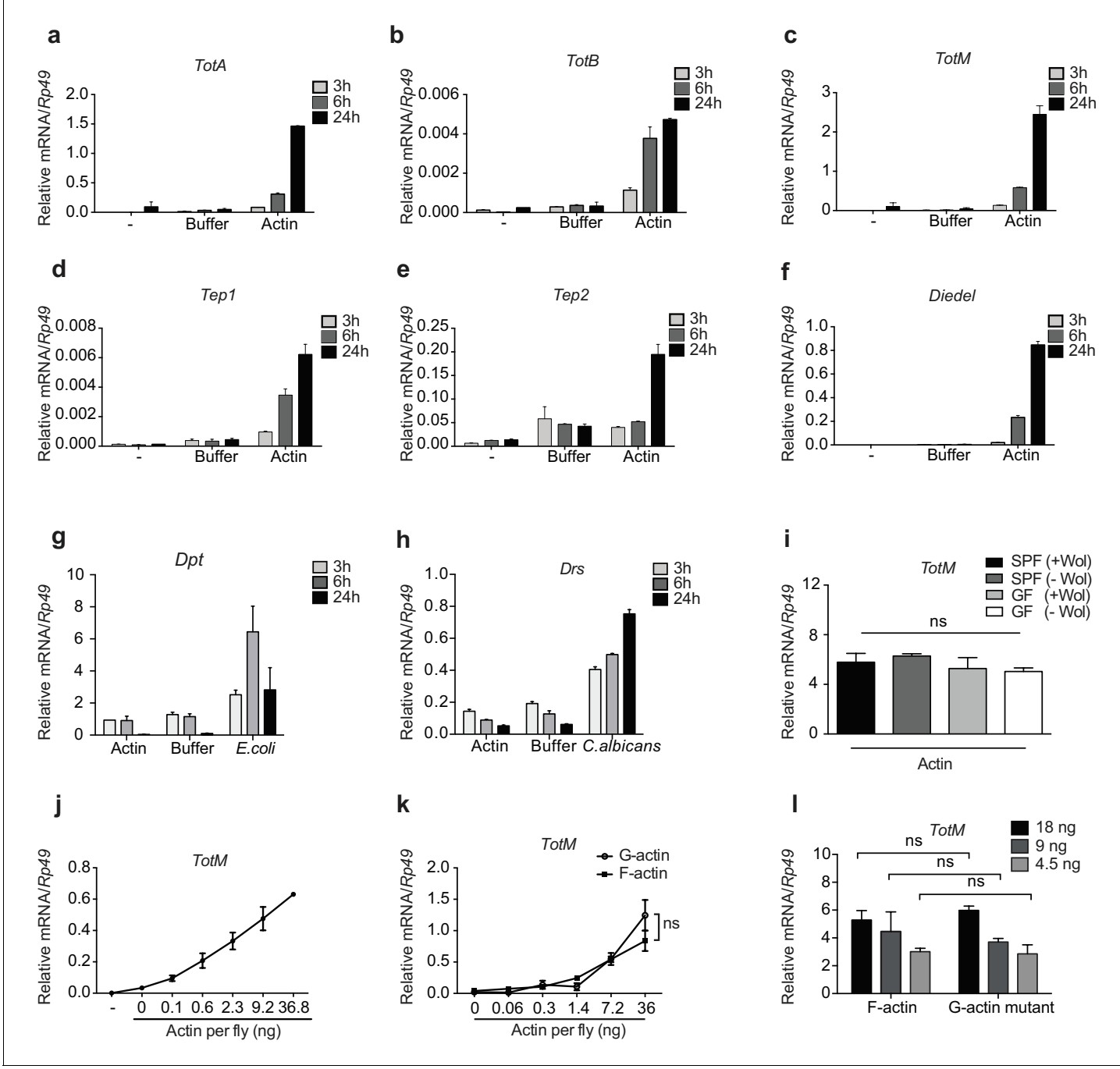

**Figure 2.** G- and F-actin induce a unique sterile inflammatory response upon injection into *Drosophila*. Control ($w^{1118}$) flies were mock treated (-) or injected with actin buffer, actin, *E. coli* or *C. albicans*, as indicated. Flies were euthanised at 3, 6 and 24 hr post injection, RNA was isolated and relative gene expression determined by quantitative RT-PCR. Depicted are expression levels of (**a–c**) *Turandot* (*Tot*) *A*, *TotB* and *TotM*, (**d–e**) *Thioester-containing protein* (*Tep*) *1* and *Tep2*, (**f**) *Diedel*, (**g–h**) *Diptericin* (*Dpt*) and *Drosomycin* (*Drs*). (**i**) Flies colonised with or free of *Wolbachia* (Wol) were reared under standard (SPF) or germ-free (GF) conditions. Relative *TotM* expression was assessed 24 hr post actin injection. (**j**) Dose-dependent *TotM* response to injected actin at 24 hr post injection. (**k**) Dose-response curve for latrunculin B-stabilised G-actin vs phalloidin-stabilised F-actin diluted in G-actin or F-actin buffer, respectively. Relative expression levels of *TotM* 24 hr post injection are shown. (**l**) A non-polymerisable G-actin mutant and F-actin were serially diluted in F-buffer at the indicated concentrations before injection into flies. Relative expression levels of *TotM* 24 hr after injection are depicted. Relative gene expression levels were calculated using the housekeeping gene *Rp49* as a reference gene. Data are representative of at least two independent experiments with 10 flies/sample with duplicate samples. Bars represent mean ± SEM. Statistical analysis was performed using two-way ANOVA with Sidak's multiple comparison test as post-test for pairwise comparisons. Results of Sidak's multiple comparison test are shown (ns, *Figure 2 continued on next page*

*Figure 2 continued*

not significant; *p<0.05; **p<0.01; ***p<0.001; ****p<0.0001). If not otherwise indicated, flies were injected with 36.8 ng of actin per fly. All data points are plotted even where no bars are visible.

The following figure supplements are available for figure 2:

**Figure supplement 1.** G-actin and F-actin induced *TotM* peaks after 24 hr.

**Figure supplement 2.** *TotM* is selectively induced by injected actin.

extracts. First, we validated the requirement for the JAK/STAT pathway by expressing a dominant negative Domeless receptor (*dome ΔCyt2.3*). Overexpression of *dome ΔCyt2.3* in the whole fly using a ubiquitous Actin-Gal4 driver resulted in significant attenuation of *TotM* expression in response to actin injection (*Figure 4a*). The same result was obtained when overexpressing *dome ΔCyt2.3* specifically in the fat body using a fat body-restricted, temperature-sensitive c564-Gal80^ts driver (c564-Gal4; *Tub*-Gal80^ts) (*Figure 4a*). Concordant with those results, the response was also attenuated in extracts of flies in which STAT signalling was reduced by overexpressing the STAT inhibitor, dPIAS, in the fat body using two different fat body-specific drivers under the control or not of a Gal80^ts temperature-sensitive repressor (*Figure 4b, c*). Finally, RNAi-mediated knockdown of STAT92E specifically in fat body cells was sufficient to suppress *TotM* induction by injected actin (*Figure 4d*). These results suggest that canonical STAT-dependent *TotM* induction in the fat body accounts for the global response seen in total fly extracts.

Although we focused on *TotM*, we also assessed whether some of the other most highly upregulated genes in actin-injected flies were similarly STAT-dependent. However, ablation of STAT92E in the fat body did not alter the inducibility of those we tested (*Figure 4—figure supplement 1a*). In contrast, actin-induced downregulation of some tested transcripts was prevented by reduction of STAT92E in the fat body (*Figure 4—figure supplement 1b*). These data, together with the GSEA analysis, underscore the notion that STAT activation is a major outcome of actin injection but suggest that extracellular actin likely triggers additional signalling pathways, which may or not involve the fat body and impact gene expression and/or transcript stability. Alternatively, the residual STAT92E protein in c564-Gal80^ts>UAS-*STAT92E* IR flies might be sufficient for inducing expression of certain genes that require a lower threshold of STAT activity.

## Extracellular actin-induced STAT activation involves a Upd3 paracrine cytokine loop in the fat body

Upd3 produced by haemocytes is essential to induce STAT responses in the fat body or in the intestine of flies subjected to septic injury or a high fat diet (*Agaisse et al., 2003*; *Chakrabarti et al., 2016*; *Woodcock et al., 2015*). To determine if haemocytes were similarly required for the fat body STAT response to actin, we genetically ablated them in adult flies using two haemocyte-specific (*croquemort* or *Hemolectin*) temperature-sensitive Gal80^ts-Gal4 drivers crossed to a UAS-*rpr* strain encoding Reaper, a protein that induces apoptosis. Ablation of haemocytes in flies containing UAS-*rpr*, but not in control flies, was confirmed by confocal microscopy taking advantage of a fluorescent reporter protein to mark the cells (*Figure 5a, b*). Despite the near-complete elimination of haemocytes, *TotM* induction in response to actin in UAS-*rpr* flies was indistinguishable from that in controls (*Figure 5c*), indicating that haemocytes are redundant. We further established that haemocyte-derived Upd3 is dispensable by using a *upd3* RNAi line crossed to another line carrying a *Hemolectin*-Gal4 driver (*Figure 5d*). We therefore tested the possibility that Upd3 might be essential but made by fat body cells themselves rather than haemocytes. Consistent with that notion, loss of the *TotM* response to actin injection was seen when the *upd3* RNAi line was crossed to a line bearing a r4-Gal4 fat body driver (*Figure 5d*). Similar results were obtained using a different fat body driver (c564-Gal4), this time under the control of Gal80^ts (*Figure 5e*). In contrast to Upd3, knockdown of Upd1 and Upd2 in fat body cells had no effect on the response to actin (*Figure 5e*), confirming Upd3 as the key cytokine. We conclude that extracellular actin leads to Upd3 production by fat

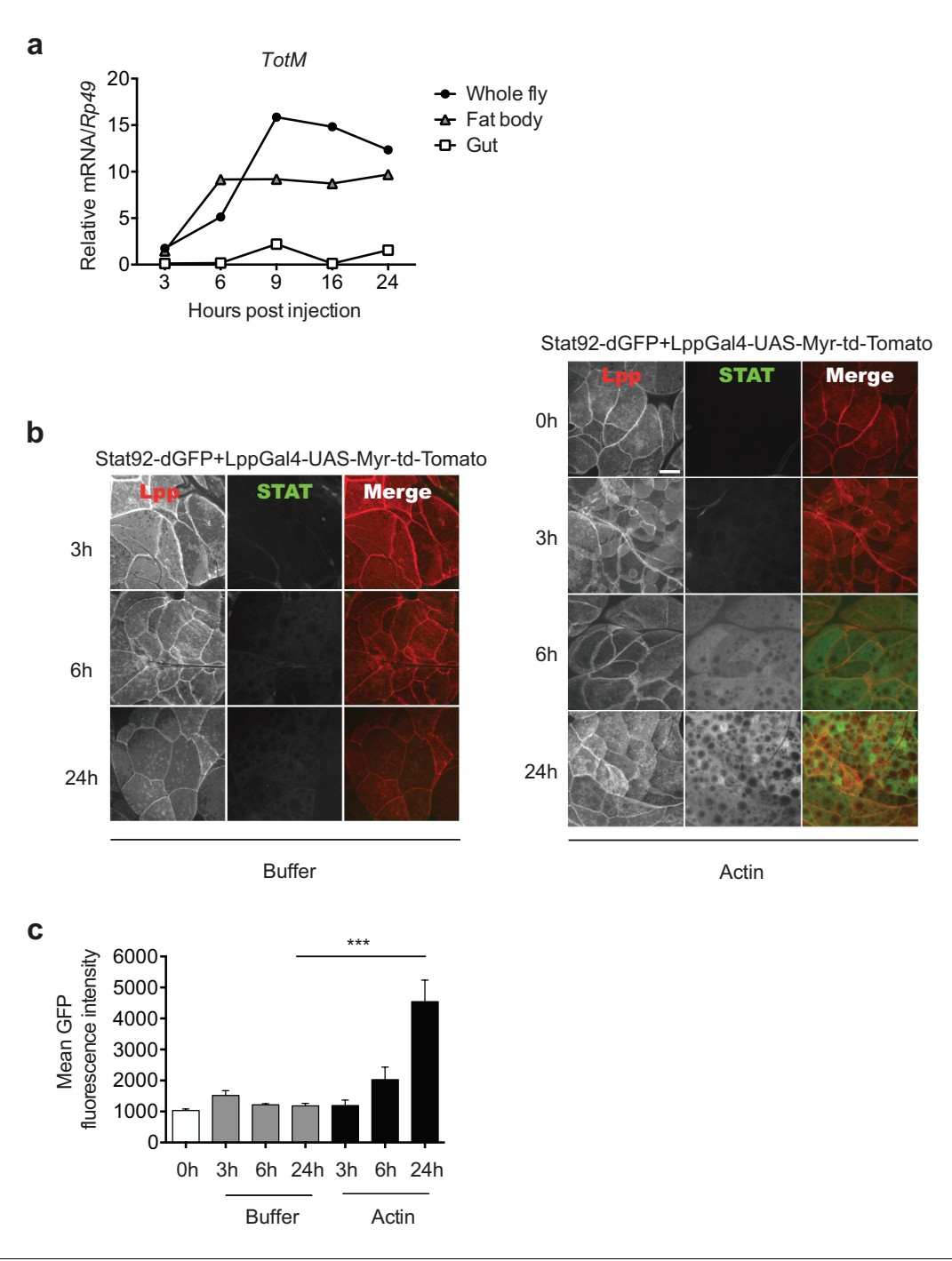

**Figure 3.** Actin injection induces JAK/STAT activation in the fat body. (a) Control ($w^{1118}$) flies were injected with actin and euthanised at the indicated times, after which the fat bodies and intestines were dissected and RNA extracted. Relative *TotM* expression in the two organs compared to whole flies is depicted. At each time point, samples represent five whole flies, 15 intestines or 15 fat bodies. *TotM* relative levels were calculated using the housekeeping gene *Rp49* as a reference gene. (b) STAT92E reporter activity in the fat bodies of Stat92-dGFP+Lpp-Gal4 > UAS-Myr-td-Tomato flies at 3, 6 and 24 hr after injection with actin (right panel) of buffer (left panel). Scale bar represents 20 μm. (c) Quantification of mean STAT fluorescence within the fat body (n = 3–11 flies). Bars represent mean ± SEM. Statistical analysis was performed using one-way ANOVA with Sidak's multiple comparison test as post-test for pairwise comparisons. Results of Sidak's multiple comparison test are shown (ns, not significant; *p<0.05; **p<0.01; ***p<0.001; ****p<0.0001). All data points are plotted even where no bars are visible.

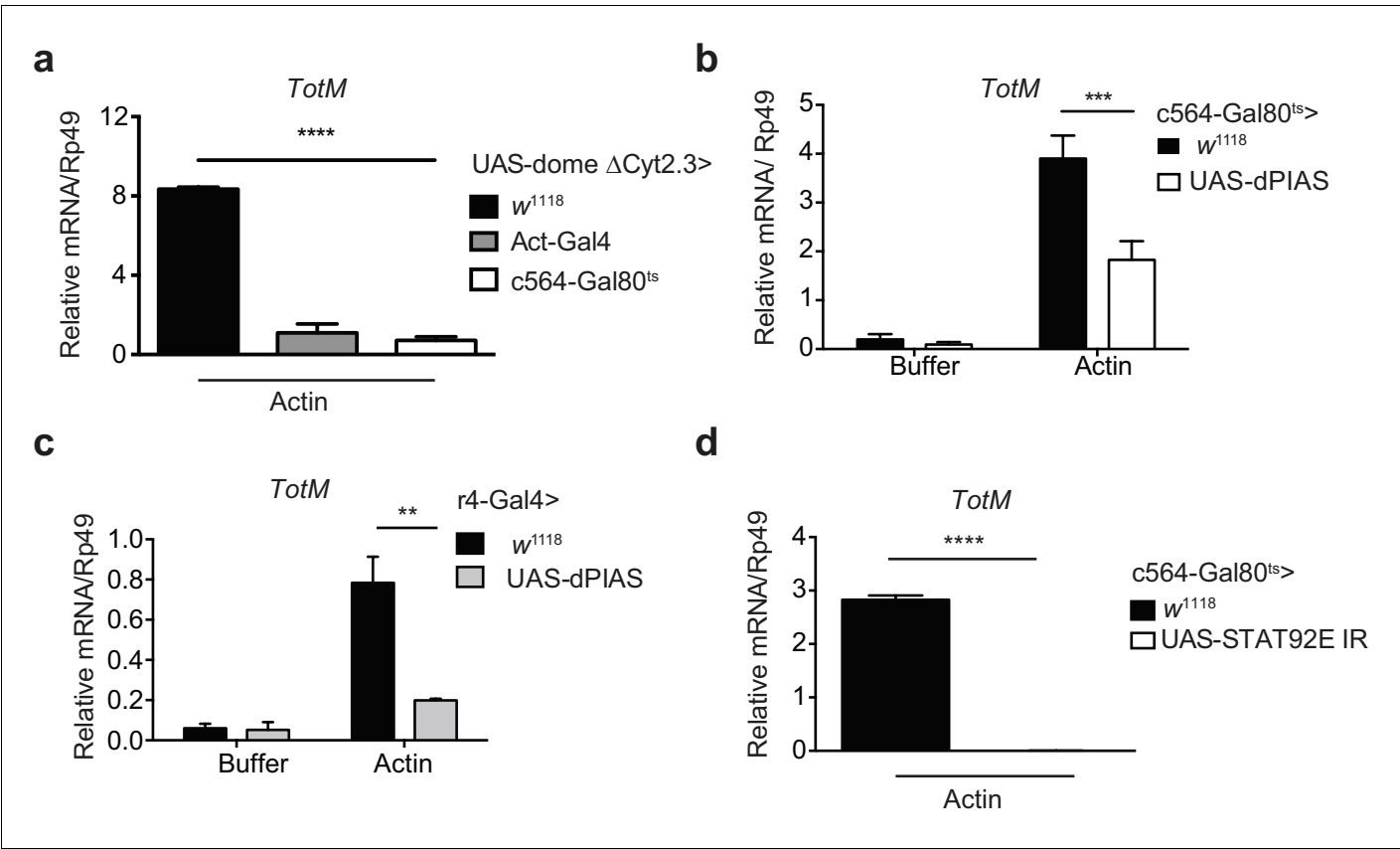

**Figure 4.** Extracellular actin-driven *TotM* expression depends on canonical Domeless signalling in the fat body. (a) Relative *TotM* expression in flies overexpressing *domeless ΔCyt2.3* dominant-negative isoform or no transgene (*w*[1118] control) under the control of a ubiquitous driver, *Act*-Gal4, or a fat body-inducible driver c564-Gal4; *Tubulin*-Gal80[ts] (c564-Gal80[ts]), 24 hr after injection with actin. (b) Relative *TotM* expression in flies overexpressing *dPIAS* under the control of the fat body inducible driver c564-Gal4; *Tubulin*-Gal80[ts] (c564-Gal80[ts]), 24 hr after injection with either buffer or actin. Data are pooled from two independent experiments with 5–10 flies/sample and duplicate samples. (c) Relative *TotM* expression in flies overexpressing *dPIAS* under the control of the fat body constitutive driver r4-Gal4, 24 hr after injection with either buffer or actin. Data are representative of three independent experiments with 5–10 flies/sample with duplicate samples. (d) Relative *TotM* expression in flies overexpressing UAS-*STAT92E* IR under the control of the fat body inducible driver c564-Gal4; *Tubulin*-Gal80[ts] (c564-Gal80[ts]), 24 hr after injection with actin. Data are pooled from two independent experiments with triplicate samples and 5–10 flies/sample. *TotM* relative levels were calculated using the housekeeping gene *Rp49* as a reference gene. Bars represent mean ± SEM. Statistical analysis was performed using two-way ANOVA with Sidak's multiple comparison test as post-test for pairwise comparisons (a, b, c) or unpaired t-test (d). Significant differences with Sidak's multiple comparison test or unpaired t-test are shown (ns, not significant; *$p<0.05$; **$p<0.01$; ***$p<0.001$; ****$p<0.0001$). All data points are plotted even where no bars are visible.

The following figure supplement is available for figure 4:

**Figure supplement 1.** STAT-dependence of selected actin-induced or -repressed genes.

body cells, which acts in an autocrine or paracrine fashion via Domeless to induce STAT activation and induction of STAT response genes.

## Response to extracellular actin requires activation of Src42A and Shark in the fat body

There is no DNGR-1 orthologue in *Drosophila*. Furthermore, the ability of G-actin to trigger *TotM* suggested that a functionally-equivalent receptor is not involved. However, it was conceivable that signalling pathways downstream of DNGR-1 activation by F-actin might also be employed in the fruit fly response to extracellular G-actin. We therefore tested the requirement for Shark, a non-receptor tandem SH$_2$ kinase and *Drosophila* Syk orthologue (*Fernandez et al., 2000*). Indeed, conditional silencing of *Shark* in adult flies using a ubiquitous driver led to a significant reduction in

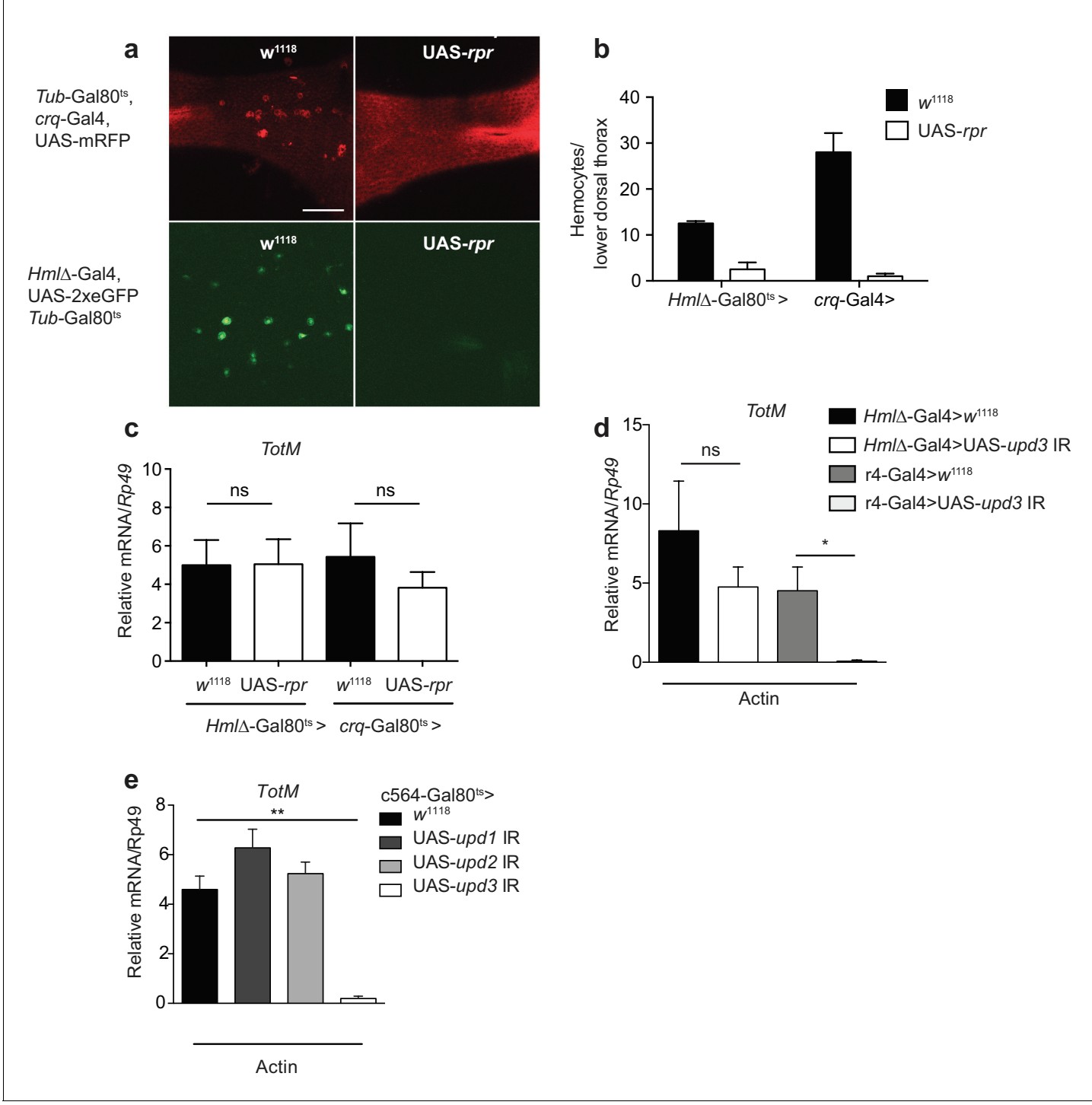

**Figure 5.** Extracellular actin-driven *TotM* expression requires fat body-derived Upd3 but not haemocytes. (a) Intravital confocal microscopy of the dorsum of *Tub*-Gal80ts, *crq*-Gal4, UAS-mRFP or *Hml*△-Gal4, UAS 2xeGFP; *Tub*-Gal80ts lines crossed to UAS-*rpr* or control (*w*1118) flies after shifting to 29°C. (b) Quantification of the images shown in (a). Haemocyte numbers in the lower dorsal thorax counted from two images (*Hml*△-Gal4) or four images (*crq*-Gal4). Differences in overall haemocyte numbers between the two driver lines may be due to differing specificities of the *crq* and *Hml* markers. (c) Relative *TotM* expression in haemocyte-deficient (*Hml*△-*Gal4* > UAS-*rpr* or *crq*-Gal4 > UAS-*rpr*) vs control flies (*Hml*△*Gal4* > *w*1118 or *crq*-Gal4 > *w*1118), 24 hr after injection with actin. In order to increase statistical power, data were pooled from two independent experiments with 5–10 flies/sample and triplicate samples. (d) *TotM* expression levels in the constitutive fat body driver (*r4*-Gal4) or constitutive haemocyte driver lines (*Hml*△-Gal4) crossed to either control (*w*1118) or UAS-*upd3* IR lines, 24 hr after injection with actin. Data are representative of two independent experiments with 5–10 flies/sample and triplicate samples. (e) *TotM* expression levels in fat body driver line c564-Gal4; *Tubulin*-Gal80ts (c564-Gal80ts) crossed to

*Figure 5 continued on next page*

Figure 5 continued

control ($w^{1118}$), UAS-*upd1* IR, UAS-*upd2* IR or UAS-*upd3* IR lines shifted to the restrictive temperature, 24 hr after injection with actin. Data are representative of three independent experiments with 5–10 flies/sample and triplicate samples. *TotM* relative levels were calculated using the housekeeping gene *Rp49* as a reference gene. Bars represent mean ± SEM. Statistical analysis was performed using one-way ANOVA with Sidak's multiple comparison test as post-test for pairwise comparisons. Results of Sidak's multiple comparison test are shown (ns, not significant; *$p<0.05$; **$p<0.01$; ***$p<0.001$; ****$p<0.0001$). All data points are plotted even where no bars are visible.

*TotM* expression upon actin injection (*Figure 6a*). Next, we asked whether the fat body itself might be the primary site of Shark signalling. Consistent with that notion, knockdown of *Shark* within the fat body using a c564-Gal80$^{ts}$ driver led to a large reduction in the levels of actin-induced *TotM* in total flies that had been shifted to the restrictive temperature (*Figure 6b*). This result was confirmed using a different driver (r4-Gal4) that led to constitutively reduced Shark levels in the fat body (*Figure 6c*). Shark activation occurs downstream of Src family kinase activity (*Ziegenfuss et al., 2008*; *Evans et al., 2015*). Silencing of the Src family kinase *Src42A* using a ubiquitous temperature-sensitive driver also led to a striking reduction in actin-induced *TotM* levels (*Figure 6d*). As for Shark, Src42A was specifically required in the fat body because silencing with fat body-specific drivers also led to complete loss of actin-induced *TotM* (*Figure 6d*). In keeping with the knockdown data, overexpression of a dominant negative allele of *Src42A* (*Src42A*$^{DN}$) using an ubiquitous or fat body-specific driver similarly led to a marked reduction in actin-driven *TotM* levels (*Figure 6e*). Finally, the overexpression of a constitutively active form of Src42A within the fat body was sufficient to induce a *TotM* response that was comparable in magnitude to that induced by actin injection (*Figure 6f*).

Uptake of axonal debris by glial cells and responses to wound healing in *Drosophila* both require Src42A phosphorylation of the ITAM-bearing receptor Draper (*drpr*), which serves as a platform for Shark recruitment (*Ziegenfuss et al., 2008*; *Evans et al., 2015*). We found that *drpr* null mutants displayed a reduced *TotM* response to actin injection (*Figure 6—figure supplement 1a*). However, this reduction was not specific to Draper loss as it could not be rescued by complementation (*Figure 6—figure supplement 1b, c*). Consistent with that finding, conditional RNAi-mediated knockdown of *drpr* in the fat body had no effect (*Figure 6—figure supplement 1d*). We conclude that expression of Src42A and Shark in the fat body, but not of Draper, are essential for the response to extracellular actin.

## Response to extracellular actin requires Nox and generation of $H_2O_2$ in the fat body

Src family kinases are redox sensitive (*Giannoni et al., 2005*) and can be activated by wound-derived $H_2O_2$ in both zebrafish and *Drosophila* (*Yoo et al., 2011*; *Niethammer et al., 2009*; *Razzell et al., 2013*; *Evans et al., 2015*). Consistent with a requirement for superoxide in the response to extracellular actin, conditional expression of cytoplasmic superoxide dismutase (Sod) (*Missirlis et al., 2003*) ubiquitously or in the fat body diminished *TotM* induction by injected actin (*Figure 7a*). $H_2O_2$ can be generated by either of two conserved NADPH oxidases, dual oxidase (Duox) and NADPH oxidase (Nox), both of which are present as single family members in the *Drosophila melanogaster* genome (*Bae et al., 2010*). Conditional knockdown of Duox in the fat body of adult flies had no effect on the *TotM* response to injected actin (*Figure 7b*). In contrast, RNAi-mediated knockdown of Nox completely abrogated *TotM* induction, especially when using the c564 fat body-restricted driver (*Figure 7c, d*). Together, these data suggest that extracellular actin leads to Nox activity in the fat body, which causes oxidation-dependent activation of Src42A and phosphorylation and activation of downstream targets, including Shark.

## Septic injury is accompanied by actin release into haemolymph and Nox/Src42A-dependent TotM induction in the fat body

Previous studies have shown that septic injury of adult flies leads to JAK/STAT activation and *TotM* induction in the fat body (*Brun et al., 2006*; *Agaisse et al., 2003*). Using a model of septic injury with either *Escherichia coli* or *Micrococcus luteus*, we found rapid accumulation of actin in the haemolymph of infected flies but not mock (uninfected) controls (*Figure 8a*). As previously described,

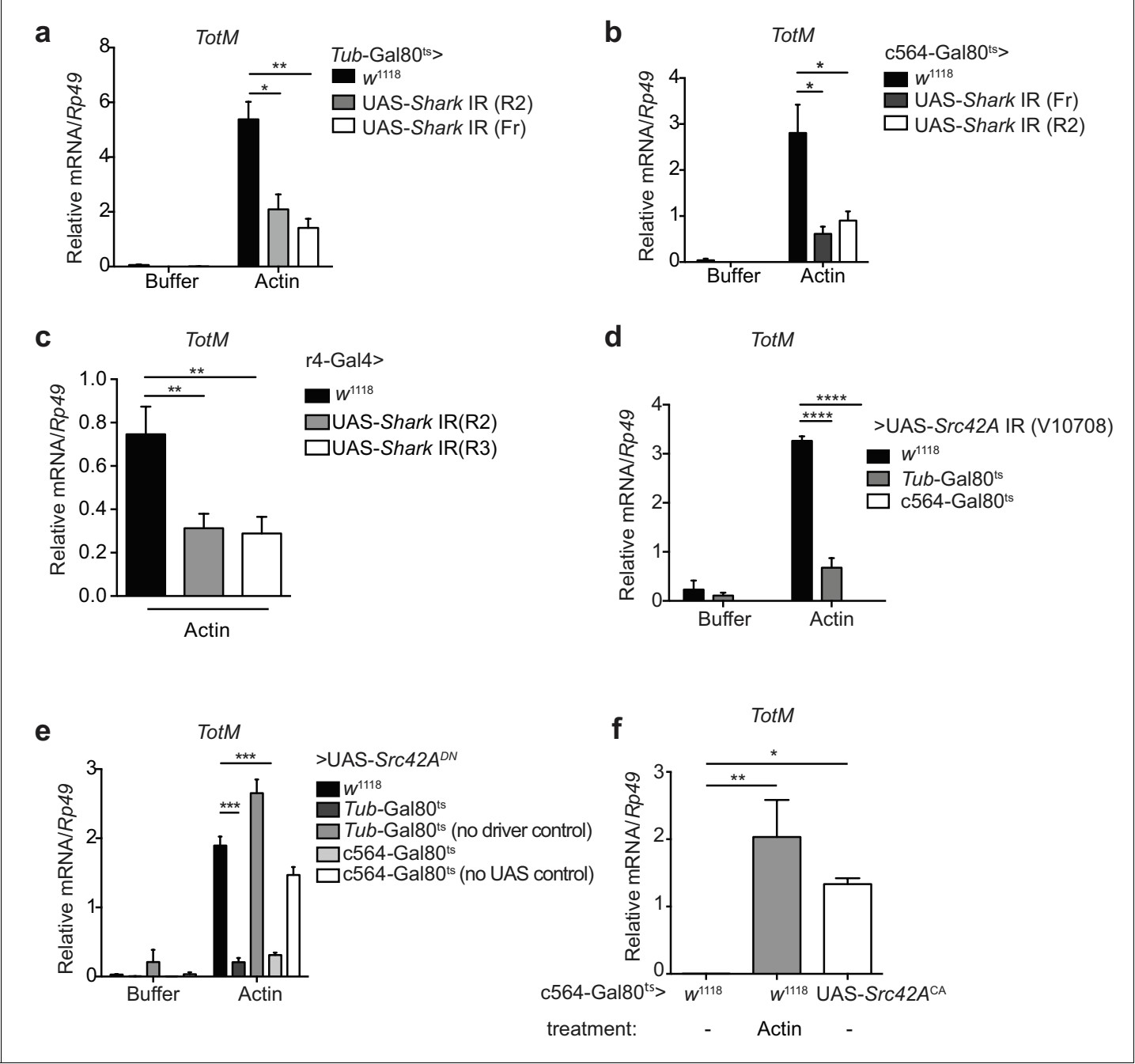

**Figure 6.** Extracellular actin-induced TotM expression requires *Shark* and *Src42A* in the fat body. (**a**)*TotM* expression levels in flies that lack *Shark* ubiquitously (*Tub*-Gal80ts; *Tub*-Gal4 > UAS-*Shark* IR R2/Fr) or in control flies lacking a driver (*w*[1118] > UAS-*Shark* IR R2/Fr), 24 hr after injection with either buffer or actin. Data are representative of three independent experiments with 5–10 flies/sample with duplicate samples. (**b**) Flies lacking *Shark* selectively in the fat body (c564-Gal80ts > UAS-*Shark* IR R2/Fr) or control flies lacking a driver (*w*[1118] > UAS-*Shark* IR R2/Fr) were injected with either buffer or actin. Relative *TotM* levels 24 hr post injection are depicted. Data are representative of two independent experiments with 5–10 flies/sample and triplicate samples. (**c**) The relative *TotM* expression in a constitutive fat body driver line (r4-Gal4) crossed to control (*w*[1118]) or UAS-*Shark* IR (R2/R3), 24 hr after injection with actin. Data are representative of three independent experiments with 5–10 flies/sample and duplicate samples. (**d**) Relative *TotM* levels in flies lacking *Src42A* either ubiquitously (*Tub*-Gal80ts > UAS-*Src42A* IR) or selectively in the fat body (c564-Gal80ts > UAS-*Src42A* IR) compared to control flies lacking a driver (*w*[1118] > UAS-*Src42A* IR), 24 hr after injection with either buffer or actin. Data are representative of three independent experiments with 5–10 flies/sample with duplicate samples. (**e**) Relative *TotM* levels in flies expressing a dominant negative version of *Src42A* ubiquitously (*Tub*-Gal80ts > UAS-*Src42A*[DN]), within the fat body (c564-Gal80ts > UAS-*Src42A*[DN]) or in the absence of a driver (*w*[1118] > UAS-*Src42A*[DN]), 24 hr after injection with either buffer or actin. No driver control refers to *Tub*-Gal80ts; UAS-*Src42A*[DN]/TM6C.Sb[1]. No UAS control refers to c564-Gal4; *Tub*-Gal80ts/TM6C.Sb[1]. Data are representative of two independent experiments with 5–10 flies/sample with triplicate samples. (**f**) Relative

*Figure 6 continued on next page*

*Figure 6 continued*

*TotM* levels between untreated (-) fat body driver line crossed to constitutively active Src42A (c564-Gal80$^{ts}$ > UAS-*Src42A*$^{CA}$); untreated control flies (c564-Gal80$^{ts}$ > $w^{1118}$) and actin-injected control flies (c564-Gal80$^{ts}$ > $w^{1118}$). Data are representative of three independent experiments with 5–10 flies/ sample with triplicate samples. *TotM* relative levels were calculated using the housekeeping gene *Rp49* as a reference gene. Bars represent mean ± SEM. Statistical analysis was performed using one-way (c, f) or two-way (a, b, d, e) ANOVA with Sidak's multiple comparison test as post-test for pairwise comparisons. Results of Sidak's multiple comparison test are shown (ns, not significant; *p<0.05; **p<0.01; ***p<0.001; ****p<0.0001). All data points are plotted even where no bars are visible.

The following figure supplement is available for figure 6:

**Figure supplement 1.** Extracellular actin-induced *TotM* expression is independent of Draper.

septic injury was accompanied by the induction of *TotM* (**Figure 8b**). Strikingly, knockdown of *Nox* in the fat body led to roughly a 40% reduction in *TotM* while knockdown of *Src42A* completely abrogated the response (**Figure 8b**). Importantly, expression of the Toll- or and Imd-regulated AMPs, *Dpt* and *Drs*, was unaltered by *Nox* or *Src42A* knockdown in the fat body (**Figure 8c, d**). These data are consistent with the possibility that actin released into the haemolymph after septic injury can trigger JAK/STAT activation in the fat body via the Nox/Src42A-dependent pathway described here.

## Discussion

Inflammation is a response to microbial invasion or tissue damage designed to eliminate the offending stimulus, clear debris and stimulate tissue repair. While we have learned much about the pathways that trigger inflammation in response to pathogen invasion (*Medzhitov, 2010*; *Takeuchi and Akira, 2010*), we still understand relatively little about induction of sterile inflammation following tissue injury (*Rock et al., 2010*). Importantly, dysregulated and/or chronic inflammation, often of sterile origin, is increasingly recognised as a contributing factor to a vast range of human diseases, from cancer to neurodegeneration (*Okin and Medzhitov, 2012*; *Rock and Kono, 2008*; *Heneka et al., 2015*; *Zelenay and Reis e Sousa, 2013*; *Hanahan and Weinberg, 2011*). Furthermore, because injury and infection often overlap, our understanding of immunity necessitates a consideration of the interplay between the processes that detect pathogen invasion and those that sense tissue damage. The study of invertebrate responses to DAMPs might therefore lead to a new understanding of sterile inflammation and the identification of conserved eliciters, detectors and signaling pathways that are utilised across evolution to detect loss of cell integrity.

We and others have previously reported that actin, one of the most abundant and conserved proteins in eukaryotic cells, acts as a DAMP in mouse and humans, binding to DNGR-1, a Src and Syk-coupled dead cell receptor expressed on DCs (*Ahrens et al., 2012*; *Sancho et al., 2009*; *Zhang et al., 2012*; *Hanč et al., 2015*). Here, we provide evidence that actin is also a DAMP in *Drosophila melanogaster*, triggering a response that, like in vertebrates, requires Syk and Src family kinases. We show that the presence of extracellular actin in the haemolymph of *Drosophila* elicits a reaction in the fat body via Shark and Src42A, whose activation depends on reactive oxygen species (ROS) generated by the NADPH oxidase Nox. Consistent with our data, ROS generation by NADPH oxidases is a highly conserved response to wounding (*Nam et al., 2012*; *Takeishi et al., 2013*) and has been shown to directly activate Lyn/Src42A in zebrafish and *Drosophila* through oxidation of a single redox-sensitive cysteine residue (*Yoo et al., 2011*; *Niethammer et al., 2009*; *Evans et al., 2015*; *Razzell et al., 2013*).

In contrast to DNGR-1 dependent recognition, the fly response to extracellular actin is elicited equally by G- and F-actin, does not require phagocytes but the fat body and its function is not to prime adaptive immunity, which is absent in invertebrates. Rather, it is coupled to production of Upd3 cytokine, which acts in an autocrine and paracrine manner to induce Domeless signalling via STAT and to cause the induction of STAT-responsive genes, the products of which are released into the haemolymph. This systemic inflammatory-like response involving cytokine amplification and the fat body is reminiscent of the acute phase response in mammals, which can be triggered by infection or trauma and leads to the production of cytokines such as IL-6 that act on the liver

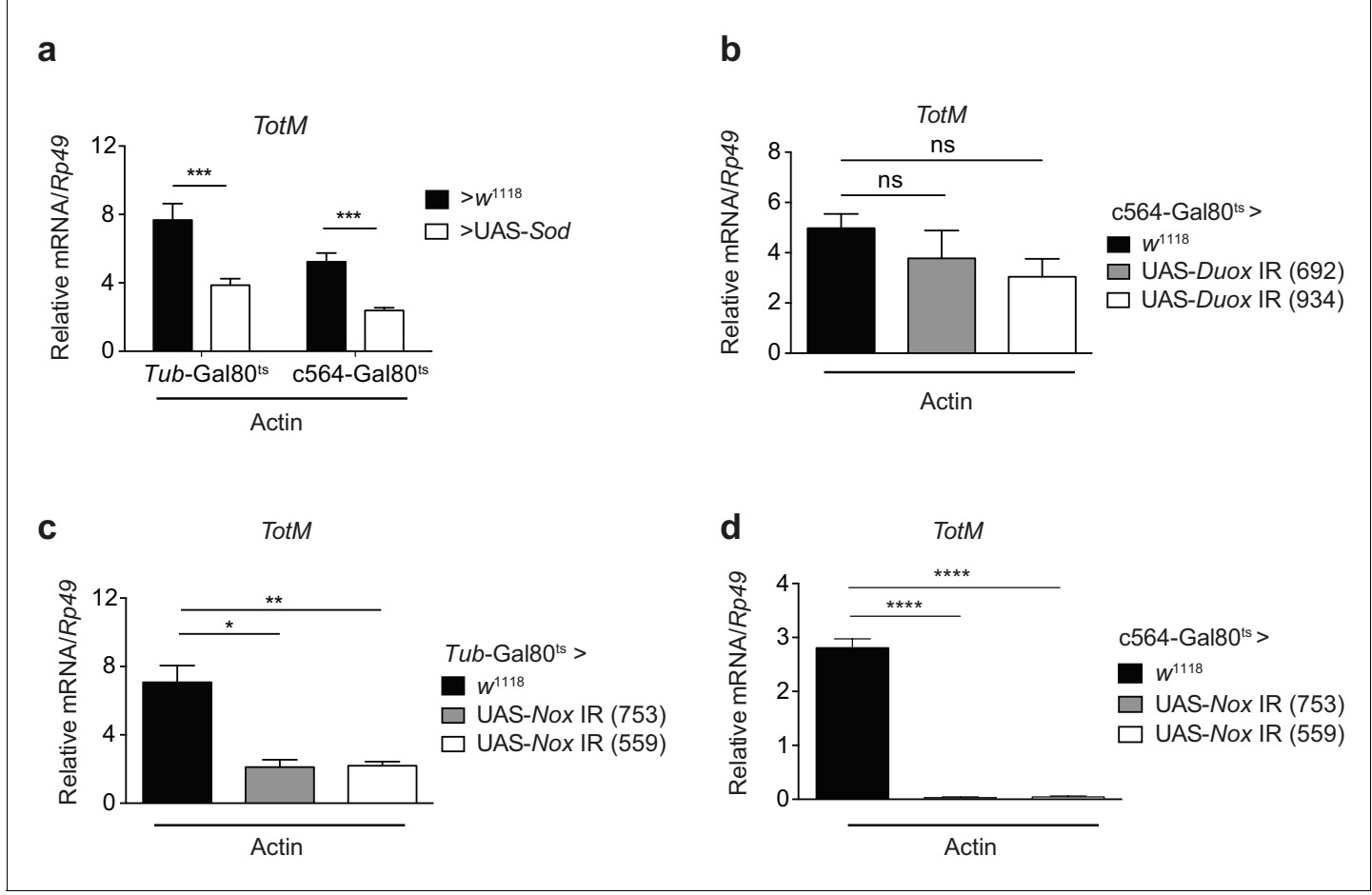

**Figure 7.** Extracellular actin-induced *TotM* expression is dependent on the NADPH oxidase Nox. (a) Flies overexpressing superoxide dismutase (Sod) either ubiquitously (*Tub*-Gal80[ts]; *Tub*-Gal4 > UAS-*Sod*) or in the fat body (*Tub*-Gal80[ts]; c564-Gal4 > UAS-*Sod*) or control flies without transgene (*Tub*-Gal80[ts]; c564-Gal4 > *w*[1118] or *Tub*-Gal80[ts]; *Tub*-Gal4 > *w*[1118]) were injected with actin. *TotM* expression 24 hr post injection is shown. In order to increase statistical power, data were pooled from two independent experiments with 5–10 flies/sample and quadruplicate samples. (b) *TotM* expression levels 24 hr post actin injection in flies in which *Duox* was knocked down in the fat body (*Tub*-Gal80[ts]; c564-Gal4 > UAS-*Duox* IR). Data are representative of two independent experiments with 5–10 flies/sample and triplicate samples. (c–d) *TotM* expression levels in flies in which *Nox* was knocked down either ubiquitously (c) (*Tub*-Gal80[ts]; *Tub*-Gal4 > UAS-*Nox* IR) or in the fat-body (d) (*Tub*-Gal80[ts]; c564-Gal4 > UAS-*Nox* IR) compared to control flies lacking RNAi (*Tub*-Gal80[ts]; *Tub*-Gal4 > *w*[1118] or *Tub*-Gal80[ts]; c564-Gal4 > *w*[1118]), 24 hr after injection with actin. Data are representative of two independent experiments with 5–10 flies/sample and triplicate samples. *TotM* relative levels were calculated using the housekeeping gene *Rp49* as a reference gene. Bars represent mean ± SEM. Statistical analysis was performed using one-way ANOVA with Sidak's multiple comparison test as post-test for pairwise comparisons. Results of Sidak's multiple comparison test are shown (ns, not significant; *p<0.05; **p<0.01; ***p<0.001; ****p<0.0001). All data points are plotted even where no bars are visible.

(mammalian equivalent of the fat body) to cause production of acute phase proteins (*Medzhitov, 2010*; *Kopf et al., 1994*). These are secreted into the plasma to regulate multiple processes such as host defence, coagulation, vascular permeability and metabolism (*Medzhitov, 2010*). Similarly, the *Drosophila* fat body response to actin results in secretion into the haemolymph of proteins that may regulate multiple aspects of fly physiology that coordinately impact resistance or tolerance to insult. However, it is important to note that while some components of the extracellular actin-sensing circuitry are conserved between flies and mammals (Shark, Src42A and ROS), others are not (DNGR1, cross-presentation, dendritic cells). These differences suggest that DAMPs can be more conserved than their receptors or the responses they evoke. This is akin to pathogen-associated molecular patterns (PAMPs) such as, for example, lipopolysaccharide (LPS), a hallmark of Gram-negative bacteria. The sensing of LPS is conserved in plants, protists and animals, but the relevant receptors and subsequent responses diverge depending on the host (*Neyen et al., 2014*).

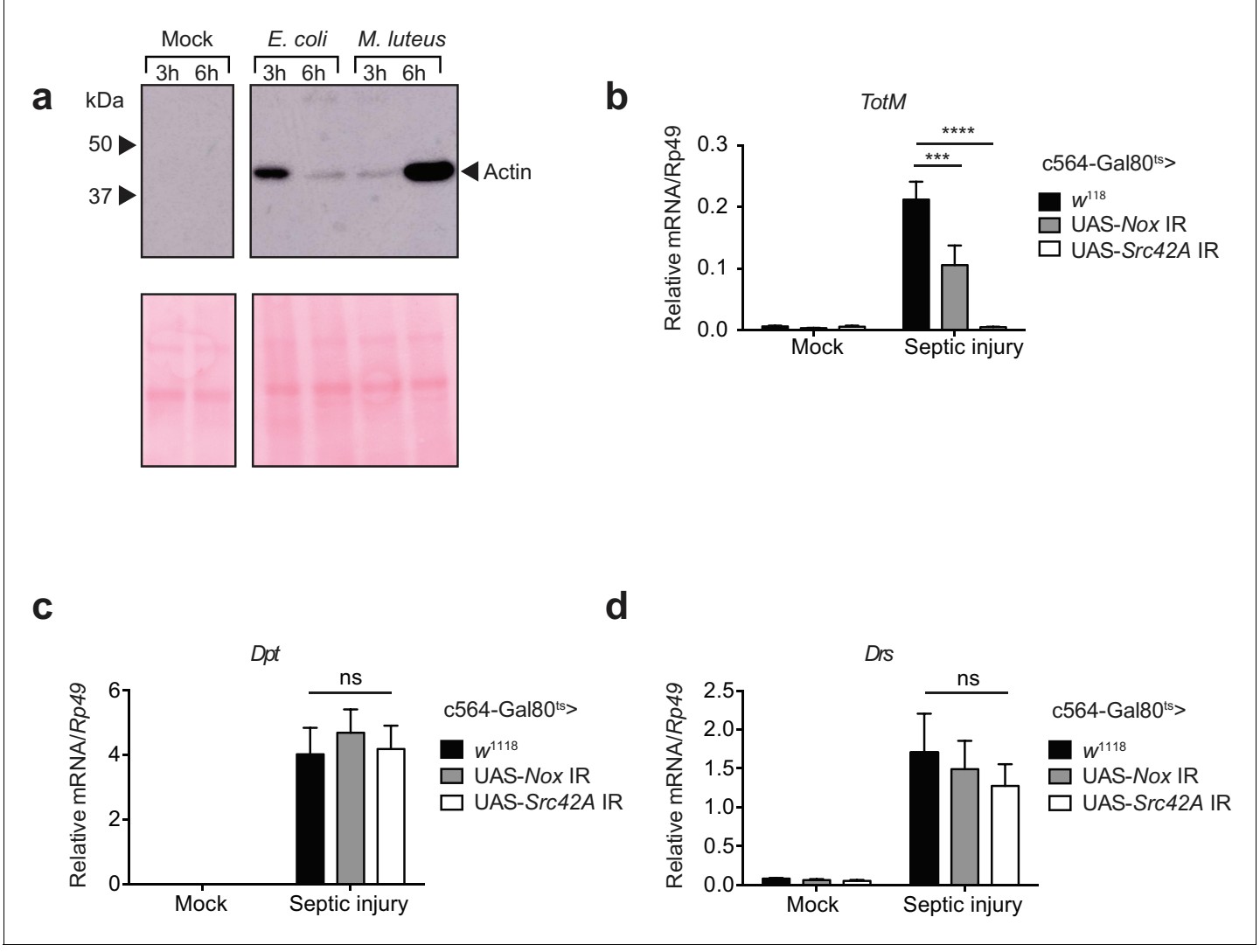

**Figure 8.** Septic injury causes actin accumulation in hemolymph and *TotM* induction dependent on fat body expression of *Nox* and *Src42A*. (a) Immunoblotting for actin in the haemolymph (Top) and Ponceau stain (bottom) of mock (uninfected), *Escherichia coli* or *Micrococcus luteus* infected flies at 3 or 6 hr post infection. Data are representative of two independent experiments with 10 flies per sample. (b–d) Expression levels of *TotM* (b), *Dpt* (c) or *Drs* (d) in flies, in which *Nox or Src42A* was knocked down in the fat-body (*Tub*-Gal80ts; c564-Gal4 > UAS-*Nox* IR or *Tub*-Gal80ts; c564-Gal4 >UAS-*Src42A* IR) compared to control flies (*Tub*-Gal80ts; c564-Gal4 > *w*[1118]), 24 hr after septic injury with a mixture of *Escherichia coli* and *Micrococcus luteus*. In order to increase statistical power, data from two independent experiments with 5–10 flies/sample and duplicate samples were pooled.

The following figure supplement is available for figure 8:

**Figure supplement 1.** Model for extracellular actin-induced JAK/STAT pathway activation.

Similarly, peptidoglycans and β-glucans are used in both flies and mammals to signify bacterial or fungal presence, yet are detected by different receptors that, nevertheless, can couple to conserved signalling pathways.

The JAK/STAT pathway in *Drosophila* can be induced by mechanical pressure, heat shock, dehydration, cytopathic infection, septic wounds and other traumas (*Ekengren et al., 2001*; *Agaisse et al., 2003*; *Pastor-Pareja et al., 2008*; *Dostert et al., 2005*). How such seemingly disparate stimuli trigger a single pathway is puzzling. However, a common denominator in all these settings is cell death and it has been speculated that STAT activation might therefore occur in response to DAMP release (*Shaukat et al., 2015*; *Buchon et al., 2014*). Our data support that

notion and suggest that actin is a potent DAMP for triggering the JAK/STAT pathway. Notably, pathogen infection in *Anopheles gambiae* and *Drosophila melanogaster* has been shown to lead to the release of actin into the haemolymph, where it can act as an antibacterial or antiparasitic agent (*Vierstraete et al., 2004*; *Sandiford et al., 2015*). Therefore, actin release may serve as a two-pronged defense mechanism, both directly as an antimicrobial and indirectly by activating a systemic JAK/STAT response.

The role of the systemic JAK/STAT response is unclear at present. Despite being commonly used as a marker of STAT activation, the function of *Tot* and *Tep* proteins in *Drosophila* is unknown. Nevertheless, genetic loss-of-function studies have implicated JAK/STAT signaling in resistance and/or tolerance to viral, bacterial and parasitoid infections (*Yang et al., 2015*; *Agaisse et al., 2003*; *Agaisse and Perrimon, 2004*; *Brun et al., 2006*; *Dostert et al., 2005*; *Lamiable et al., 2016*; *Chakrabarti et al., 2016*; *Kemp et al., 2013*; *Merkling et al., 2015*). Furthermore, the JAK/STAT pathway has a well-established role in maintenance of fly intestinal homeostasis, both at steady state and following infection or injury (*Biteau et al., 2011*; *Kohlmaier et al., 2015*; *Jiang et al., 2009*; *Micchelli and Perrimon, 2006*; *Zhou et al., 2013*; *Osman et al., 2012*; *Jiang et al., 2011*; *Beebe et al., 2010*; *Lin et al., 2010*; *Zhai et al., 2015*). Given these precedents, we attempted to investigate the role of the inducible actin-triggered JAK/STAT circuit by injecting actin into flies prior to challenge with viruses (Flock house virus, Drosophila C virus, Sindbis virus and Cricket paralysis virus) or bacteria (*Erwinia carotovora, Escherichia coli, Micrococcus luteus* and *Listeria monocytogenes*) but failed to find an effect on either resistance or tolerance to infection (data not shown). Similarly, in models of stress or injury (starvation, heat shock, irradiation, paraquat feeding and a recently-described model of concussion (*Katzenberger et al., 2013*, *2015*)), we found no evidence of protection or susceptibility afforded by actin pre-injection (data not shown). Finally, we have also not found an effect of actin injection on fat body metabolism (data not shown). The failure to find a system in which prior upregulation of STAT target genes by exogenous actin leads to a difference in outcome is a current experimental limitation. However, it might reflect the fact that STAT activation is already induced to sufficient levels in those models in response to actin released from dying cells. Consistent with this notion, we observed that septic injury led to a rapid increase in actin levels within the haemolymph. In such a situation, additional induction of the STAT pathway by actin pre-injection may not confer additional protection or tolerance. Reinforcing this notion is a recent study showing that loss of basal Diedel levels leads to reduced tolerance to Sindbis virus, yet the upregulation of Diedel levels that takes place during infection is itself dispensable (*Lamiable et al., 2016*). Unfortunately, loss-of-function experiments to assess the effect of released actin on different challenges are not feasible because actin is essential for viability. Surrogate loss-of-function experiments, such as examining the role of *Nox* and *Src42A* or *Shark* in the fat body in the context of infection or injury, have not been reported and their interpretation is complicated by the pleiotropic effects of those proteins. Nevertheless, our finding that actin is released into the haemolymph upon septic injury and that this induces JAK/STAT activation dependent on fat body expression of *Src42A* and *Nox* may suggest that previous reports of septic injury-induced STAT activation can be partially ascribed to extracellular actin.

The identity of the putative receptor that recognises extracellular actin in *Drosophila* remains unknown. The requirement for Upd3 rules out the possibility that actin serves as a direct ligand for *Domeless*, a conclusion further supported by the fact that actin does not induce *TotM* upregulation in various *Drosophila* cell lines that respond to Upd cytokines in vitro (data not shown). Therefore, the simplest interpretation of our data is that Upd3 is synthesised by fat body cells that detect extracellular actin via a sensor(s) that couple(s) to a Nox-Src42A-Shark cascade (*Figure 8—figure supplement 1*). By analogy with other receptors that engage a Syk-dependent pathway, that sensor might be an ITAM- or hemITAM-bearing receptor or one that associates in trans with an ITAM-containing signalling chain (*Brubaker et al., 2015*; *Robinson et al., 2009*). Interestingly, in *Drosophila* responses to wounding and in the clearance of axonal debris and neuronal cell corpses, one such receptor is Draper, a member of the Nimrod family and orthologue of *C. elegans* Ced1. Draper contains an ITAM that is phosphorylated by Src42A (*MacDonald et al., 2006*; *Ziegenfuss et al., 2008*; *Razzell et al., 2013*; *Evans et al., 2015*). However, we have found Draper to be dispensable for *TotM* induction in response to actin injection. Similarly, we have not found a role for Nimrod C1, C4 and the scavenger receptor CD36 (data not shown). Whether these data

indicate the activity of an unknown receptor, multiple redundant receptors or an indirect sensing mechanism, akin to the activation of the vertebrate NLRP3 receptor (**Martinon et al., 2009**), will need to be investigated.

In sum, our data suggest that extracellular actin released by dead cells induces a response in *Drosophila* that requires signalling in the fat body via the non-receptor tyrosine kinase, Shark, and the Src family kinase, Src42A. This pathway leads to production of Upd cytokines that act in an autocrine and paracrine manner to induce Domeless signalling via STAT and cause induction of STAT-responsive genes (**Figure 8—figure supplement 1**). Thus, the presence of actin in the extra-cellular space triggers a response previously associated with wounding and dead cell clearance, indicating that actin exposure acts as an ancient sign of tissue damage and that actin constitutes an evolutionarily-conserved DAMP. The notion that actin exposure can act as a universal sign of cell damage might apply more generally to other cytoskeletal proteins.

## Materials and methods

### Fly stocks

Fly stocks were raised on standard cornmeal-agar medium at 25°C. Adult female flies 3–6 days of age were used in all experiments. For heat-shock induction of transgene expression, flies were incubated for 20 min at 37°C, followed by 30 min at 18°C and another 20 min at 37°C. After the treatment, flies were allowed to recover for 6 hr at 25°C before injection. For transgene induction using the Gal80$^{ts}$ system, flies were shifted to 29°C for three days prior to injection and were kept at that temperature until euthanasia.

The following stocks were used:

| Fly stock | Description |
| --- | --- |
| $w^{1118}$ | Control strain |
| ;;UAS-*dome* △*Cyt 2.3* | Overexpression of a dominant negative form of the receptor domeless. |
| ;UAS-*dPIAS*/cyO; | Overexpression of the JAK/STAT pathway negative regulator dPIAS. |
| ;if/cyO;Hs-Gal4 | Heat shock-inducible, ubiquitous driver line. |
| $w^{1118}$;;*Hml*△-Gal4, UAS-2xeGFP, *Tub*-Gal80$^{ts}$/TM6C,Sb$^1$; | Temperature-sensitive, haemocyte-specific driver line. Haemocytes are labelled with GFP. |
| $w^{1118}$;;*crq*-Gal4, UAS-2xmRFP, *Tub*-Gal80$^{ts}$/TM6C.Sb$^1$ | Temperature-sensitive, haemocyte-specific driver line. Haemocytes are labelled with RFP. |
| w;;UAS-*rpr*/TM3.Sb$^1$ | Overexpression line for pro-apoptotic protein reaper. |
| ;;r4-Gal4/TM6C.Sb$^1$ | Constitutive fat body-specific driver line. |
| ;;Stat92-dGFP | Destabilised GFP expression under the control of a STAT response element. Bloomington stock centre number: 21699 |
| LPP-Gal4-UAS-Myr-td-Tom | Fat body line, made from LPP-Gal4 (FRT-LPPGal4-G-FRT/Tm3,Zit(Sb) from Pierre Leopold; 10XUAS-IVS-myr::tdTomato from Bloomington (BL32221). |
| $w^{1118}$;*Tub*-Gal80$^{ts}$;*Tub*-Gal4/TM6C.Sb$^1$ | Temperature-sensitive, ubiquitous driver line. |
| ;UAS-*Shark* IR; (Shark R2 $^{RNAi}$) | National Institute of Genetics stock number: 18247 R-2 |
| ;UAS-*Shark* IR; (Shark R3 $^{RNAi}$) | National Institute of Genetics stock number: 18247 R-3 |
| ;UAS-*Shark* IR; (Shark Fr $^{RNAi}$) | Interfering RNA for knockdown of shark. Kindly donated by Marc Freeman. |
| w;c564-Gal4;*Tub*-Gal80$^{ts}$ | Temperature-sensitive fat body-specific driver line. |
| ;UAS-*Src42A* IR V10708$^{RNAi}$ | Interfering RNA for knockdown of Src42A. |
| ;;UAS-*Src42A*$^{DN}$ | Overexpression of a dominant negative form of Src42A. |
| UAS-*Src42A*$^{CA}$ | Bloomington stock centre number: 6410. The DNA sequence encoding the constitutively active form of Src42A (Src42A$^{CA}$) has an amino acid substitution of Tyr$^{511}$ to Phe. Tyr$^{511}$ corresponds to the inhibitory C-terminal Tyr of Src. |
| UAS-*Nox* IR (753) | VDRC ID: 100753 |

| UAS-*Nox* IR (559) | VDRC ID: 102559 |
| UAS-*Duox* IR (692) | Bloomington stock centre number: 32903 |
| UAS-*Duox* IR (934) | Bloomington stock centre number: 33975 |
| ;;*drpr*$^{\Delta 5}$ | Null mutant for the receptor Draper. Kindly donated by Marc Freeman. |
| UAS-*drpr* IR (4833) | VDRC ID: 4833 |
| UAS-*drpr* IR (4830) | VDRC ID: 4830 |
| UAS-*drpr* IR (27084) | VDRC ID: 27084 |
| UAS-*drpr* IR (27086) | VDRC ID: 27086 |
| ;FB-Gal4; | Constitutive fat body-specific driver line. |
| UAS-*Sod* | Bloomington stock centre number: 24750 |

## Injections and infections

Unless stated otherwise, for actin injection, a 1 mg/ml purified rabbit muscle or human non-muscle G-actin (Cytoskeleton Inc., CO, USA) solution was prepared in G-actin buffer (5 mM Tris HCl [pH 8.0] +0.2 mM CaCl$_2$) as per manufacturer's instructions and 36.8 nl was administered to flies by intrathoracic injection (Nanoject II apparatus; Drummond Scientific, PA, USA). Injection of the same volume of G-actin buffer was used as a control. For experiments comparing G-actin and F-actin, human non-muscle G-actin was either stabilised with 100 µM latrunculin B or polymerised in F-actin buffer (10 mM Tris-HCl [pH 7.5]+50 mM KCl + 2 mM MgCl$_2$ and 1 mM ATP) in the presence of 5 µM phalloidin. For experiments with non-polymerisable G-actin, we used *Drosophila* 5C actin and created a Delta D-loop mutant that has residues 41–51 deleted (HQGVMVGMGQK). This mutant was expressed and purified as described (*Zahm et al., 2013*).

For other test substances, in all cases a 36.8 nl volume of each sample was injected. They included 1 mg/ml purified human HSP70 (Enzo Life Sciences, NY, USA), 1 mg/ml purified human HSP90 (Abcam, UK), 1 mg/ml monosodium urate crystals (Invivogen) and 1 mM ATP (Cytoskeleton Inc.), prepared as per manufacturer's instructions. Genomic DNA was extracted from *Drosophila*. Briefly, flies were collected and mashed in extraction buffer (10 mM Tris-HCl pH 8.2, 1 mM EDTA, 25 mM NaCl, 200 µg/ml Proteinase K). Subsequently, the sample was incubated at 37°C for 1 hr and the protease inactivated by heating to 95°C. DNA was purified using a column-based method (Qiagen, Germany). Concentration of the injected DNA was 14 ng/µl in water. KCl, CsCl, NaCl, Sucrose (Thermo Fisher Scientific, MA, USA) and Dextrose (Sigma, MO, USA) were dissolved in water and used at 100 mM. Sunflower and olive oil were from Sainsbury's and used neat. Trypsin inhibitor from bovine pancreas, heparin, chondroitin, glucagon, insulin (Sigma) and glutathione (Merck-Millipore, Germany) were prepared as 1 mg/ml solutions in PBS. Amino acids were used at a concentration of 34–500 mM in water and purchased from Sigma. For denaturation, actin was diluted in G-actin buffer supplemented with 100 mM DTT. An aliquot of that solution was denatured by heat treatment (10 min at 95°C). The heat-denatured and untreated actin aliquots were loaded into separate Dialysis Cassettes with a 3.5K molecular weight cut-off (Slide-A-Lyzer; Thermo Fisher Scientific). Dialysis was performed overnight at 4°C against PBS. Following dialysis, protein concentration in the samples was measured by BCA (Thermo Fisher Scientific) and equalised before injection.

For microbial infections, *Escherichia coli* and *Micrococcus luteus* were grown overnight in Lysogeny Broth at 37°C with shaking at 220 rpm. *Candida albicans* was grown overnight in yeast peptone dextrose media at 30°C with shaking at 220 rpm. Flies were pricked in their thorax with a needle dipped in concentrated microbial suspension (optical density of 400).

## Generation of germ-free and *Wolbachia*-free flies

Germ-free flies were generated by decontaminating fly embryos with bleach (2% sodium hypochlorite) for 10 min. Embryos were subsequently washed in ethanol (70%) for 5 min, followed by washing with sterile water. Finally the embryos were transferred using an autoclaved brush and reared on axenic food until use. *Wolbachia*-free flies were generated as previously described (*Chrostek et al., 2013*).

## Transcript analysis

Unless otherwise indicated, each experimental point was obtained by injecting or infecting one or more pools of 5–10 individual flies. Following termination of the experiment, flies were euthanised and each pool was treated as a separate sample and stored at −20°C until further use. For RNA extraction, flies in each pool were mashed using a hand-held tissue homogenizer (Kimble, TN, USA) and a small pestle (Sigma) before clarification by Qiashredder columns (Qiagen) and RNA was extracted using a column-based method with DNAse treatment (Qiagen). cDNA synthesis was performed using SuperScript II Reverse Transcriptase (Thermo Fisher Scientific), and random hexamers (Thermo Fischer Scientific). cDNA was then diluted five times in nuclease-free water and analysed for gene expression by qPCR using Express SYBR green universal master mix (Thermo Fisher Scientific). Reactions were carried out using ABI 7500 Fast or QuantStudio 7 machines (Thermo Fisher Scientific). Relative expression values were calculated from ΔCts using *Rp49* house keeping gene as a reference gene. Note that such relative expression values are not a reflection of the actual relative transcript number as they are affected by PCR efficiency and by the fact that Rp49 is expressed ubiquitously in the fly whereas many of the targets measured here (e.g., *TotM*) are only expressed in the fat body.

The following primers were utilised:

| Gene | Forward primer | Reverse primer |
| --- | --- | --- |
| *Diedel* | GTGCGTGCAATCGAAAACTA | CGTACTGCTGGTTCCTCCTC |
| *Dpt* | GCTGCGCAATCGCTTCTACT | TGGTGGAGTGGGCTTCATG |
| *Drs* | CGTGAGAACCTTTTCCAATATGATG | TCCCAGGACCACCAGCAT |
| *Rp49* | GACGCTTCAAGGGACAGTATCTG | AAACGCGGTTCTGCATGAG |
| *Tep1* | TCTGTAAAGCGGGGTGAAGT | CAGAGTAGTCAGGCCCACTT |
| *Tep2* | CCTCATGGGTGGTTACTGGT | AGCAATTACCTCACCTCGCT |
| *TotA* | GCACCCAGGAACTACTTG CATCT | GACCTCCCTGAATCGGAACTC |
| *TotB* | GGAACTCTATGCTCGGCCTA | AATCTGTCCATTCTCGCCCT |
| *TotM* | GCCAAGCCTGCACTATGAAT | GCTGTCGATGTTCCGGTATT |

## RNAseq

For each time point, three replicate pools of 10 flies each were injected with either buffer or actin. RNA was isolated from each pool before quality control checking using a Bioanalyzer and generation of independent libraries. Sequencing was performed on the Illumina HiSeq 2500 platform and generated ~53 million 100 bp paired end reads per sample. Sequenced reads were mapped to Flybase gene set (version 6.01) from BDGP6 assembly [http://flybase.org/], using RSEM (version 1.2.11) (*Li and Dewey, 2011*). RSEM uses the bowtie2 alignment tool (*Langmead and Salzberg, 2012*). Gene counts were filtered to remove genes with 10 or fewer mapped reads per sample. TMM (treated mean of M-values) normalisation and differential expression analysis using the negative binomial model was carried out with the R-Bioconductor package 'EdgeR'(*Robinson et al., 2010*) (www.bioconductor.org version 3. 1.0). Genes with logCPM > 1 and FDR < 0.05 were judged to be differentially expressed. Enrichment of fly pathways gene sets, downloaded from Fly Reactome (http://fly.reactome.org/), were assessed using GSEA (*Subramanian et al., 2005*) with logFC pre-ranked gene lists. Gene sets with an enrichment q value of less than 0.05 were judged to be statistically significant. Fastq data files are deposited in the NCBI Gene Expression Omnibus database (GSE76150).

## Promoter analysis

Genes induced by actin but not by buffer were identified and converted from Flybase IDs to Refseq IDs using ENSEMBL biomart. These genes were then interrogated using PScan (*Zambelli et al., 2009*) to identify overrepresented transcription factor binding motifs in the 500 bp region upstream

of the start site with the TRANSFAC and JASPAR databases (*Mathelier et al., 2014*; *Matys et al., 2006*).

## Haemocyte ablation

Two haemocyte drivers (*Hml*Δ-Gal4, UAS-2xeGFP, *Tub*-Gal80ts/TM6C and *crq*-Gal4, UAS-2xmRFP, *Tub*-Gal80ts/TM6C) were crossed to UAS-*rpr* for temporally controlled induction of the pro-apoptotic gene *reaper* in haemocytes. $w^{1118}$ flies were crossed to the same haemocyte driver lines as a control. Flies were grown at a permissive temperature of 18°C until adulthood and were then shifted to the non-permissive temperature of 29°C for three days to induce *rpr* expression and subsequent apoptosis of haemocytes. Flies were injected with actin or actin buffer and processed as described above. The lower dorsal thorax of at least two flies per cross was imaged using intravital confocal microscopy to assess haemocyte ablation. For this purpose, the dorsal side of the flies was affixed onto a coverslip using a drop of superglue, positioning the wings on the side. Flies were imaged within 15 min and flies that died during the procedure were excluded from the analysis. Images were acquired using an Invert LSM 710 laser scanning confocal microscope (Zeiss, Germany) equipped with a 40X Oil NA 1.25 objective and analysed with ImageJ software.

## Ex vivo imaging of fat body in dissected abdomen

The abdomens of 2–3 day old adult (;STAT92-dGFP/+, LPP-Gal4 + 10XUAS-IVS-myr::tdTomato/+) females were dissected under a dissecting microscope on a silicone dish using forceps and microscissors (Albert Heiss, Germany). The abdomen was first separated and the caudal end removed. The remainder was cut along one of the sides, placed in a drop of PBS and the gut was removed. Finally, the preparation was transferred onto a fresh drop of PBS on a glass bottom dish (MatTek Corporation P35G-0–10-C) with the exoskeleton side up and covered with a round coverglass (VWR, PA, USA Cat. No 631–0150). All imaging was carried out at room temperature. Z stacks of EGFP or RFP-expressing fat body cells in the abdomen with a 0.5 µm step size were taken using a 40×oil immersion objective lens on a Perkin Elmer UltraView spinning disc system. Images were captured using Volocity (Perkin Elmer, Waltham, MA, USA) and this set-up was also used to quantify STAT activation. The Z-stacks were cropped to the proximal half of fat body cells (4–6 slices with a 0.5 µm step size) and were assembled into maximum projections with identical adjustments made to contrast across experimental groups. Regions of interest within the cytoplasm of fat body cells were selected and the mean intensity of the GFP channel was measured.

## Haemolymph isolation and western blotting

Haemolymph was isolated by gently pricking flies in the thorax area using a 27G needle. 10 flies per group were then rapidly transferred into a pre-chilled 0.5 ml microfuge tube (modified with a small hole at the nib created using a 25G needle). This was then placed into a 1.5 ml microfuge collection tube and spun for 5 min at 5000 rpm at 4°C. The resulting small drop of haemolymph was diluted in 10 µl RIPA buffer (supplemented with protease inhibitors (Roche, Switzerland)). For Western blot, haemolymph samples were diluted into Laemmli buffer, resolved using reducing SDS-PAGE and transferred to nitrocellulose membranes (Merck-Millipore). Equivalent protein levels across samples were confirmed by membrane staining with Ponceau Red dye. Actin levels were assessed by probing membranes with HRP-linked monoclonal rabbit anti β-actin (clone 13E5, Cell Signaling Technology, CA, USA). The signal was revealed with the SuperSignal West Pico Chemiluminescent substrate kit (Thermo Fisher Scientific). Detection of F- and total actin by dot blot was carried out as described (*Ahrens et al., 2012*; *Hanč et al., 2015*).

## Acknowledgements

We thank Nic Tapon, Paul Martin, Maxine Holder, Georgina Fletcher, Ieva Gailite and members of the Immunobiology Laboratory for helpful discussions and suggestions. We thank the Francis Crick Institute Genomics Equipment Park, Advanced Sequencing Facility, and the Fly facility for assistance. We also thank the Bloomington Stock Centre, the NIG-Fly Stock Center, the Vienna Drosophila Resource Center, and Flybase. This work was supported by The Francis Crick Institute, which receives core funding from Cancer Research UK (FC001136), the UK Medical Research Council

(FC001136), and the Wellcome Trust (FC001136), and by Investigator Award WT106973MA from the Wellcome Trust.

## Additional information

### Funding

| Funder | Grant reference number | Author |
|--------|------------------------|--------|
| Wellcome | WT106973MA | Caetano Reis e Sousa |
| Wellcome | FC001136 | Caetano Reis e Sousa |
| Medical Research Council | FC001136 | Caetano Reis e Sousa |
| Cancer Research UK | FC001136 | Caetano Reis e Sousa |

The funders had no role in study design, data collection and interpretation, or the decision to submit the work for publication.

### Author contributions

NS, OG, SA, Conception and design, Acquisition of data, Analysis and interpretation of data, Drafting or revising the article; AF, Assistance with experiments, Acquisition of data ; SD, Assistance with experiments, Acquisition of data; PC, Assistance with bioinformatics analysis; DP, Assistance with experiments, Acquisition of data; AAY, MKR, RSV, LT, Provided key reagents, Contributed unpublished essential data or reagents; BT, Conception and design, Analysis and interpretation of data ; MSD, Conception and design, Analysis and interpretation of data; CReS, Conception and design, Analysis and interpretation of data, Drafting or revising the article

### Author ORCIDs

Oliver Gordon, http://orcid.org/0000-0002-2567-2482
Michael K Rosen, http://orcid.org/0000-0002-0775-7917
Luis Teixeira, http://orcid.org/0000-0001-8326-6645
Barry Thompson, http://orcid.org/0000-0002-0103-040X
Marc S Dionne, http://orcid.org/0000-0002-8283-1750
Caetano Reis e Sousa, http://orcid.org/0000-0001-7392-2119

## Additional files

### Major datasets

The following dataset was generated:

| Author(s) | Year | Dataset title | Dataset URL | Database, license, and accessibility information |
|-----------|------|---------------|-------------|--------------------------------------------------|
| Probir Chakravarty, Naren Srinivasan | 2016 | Genome wide responses to extracellular actin | https://www.ncbi.nlm.nih.gov/geo/query/acc.cgi?acc=GSE76150 | Publicly available at Gene Expression omnibus(accession no: GSE76150) |

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
