## [Decision Letter]

Thank you for submitting your article "Actin is an evolutionarily-conserved damage-associated molecular pattern that signals tissue injury in *Drosophila*" for consideration by *eLife*. Your article has been favorably evaluated by Tadatsugu Taniguchi as the Senior Editor and three reviewers, including Zhijian J Chen (Reviewer #1) – who is a member of our Board of Reviewing Editors – and Jonathan Kagan (Reviewer #2).

The reviewers have discussed the reviews with one another and the Reviewing Editor has drafted this decision to help you prepare a revised submission.

Summary:

Previous studies in the mammalian system have shown that F-actin is a ligand released from necrotic cells that binds to and activates DNGR1 in dendritic cells to promote cross presentation of dead cell-associated antigens. Expanding on these studies, Srinivasan et al. tested the hypothesis that actin is an evolutionarily conserved damage associated molecular pattern (DAMP) molecule using *Drosophila* as a model system. Indeed, they found that injection of actin into *Drosophila* triggered induction of STAT-dependent genes. They further showed that actin activates the canonical JAK-STAT pathway in the fat body of *Drosophila* through the cytokine Upd3 and the cytokine receptor Doomless. Analogous to the actin-DNGR1 pathway in mammals, the actin signaling pathway in *Drosophila* requires non-receptor tyrosine kinases Shark and Src42A, which are *Drosophila* orthologues of the mammalian kinases Syk and Src, respectively. The *Drosophila* NADPH oxidase, NOX, which generates H_2_O_2_ known to activate the Src family kinases during wound healing process, is also required for actin to induce STAT-dependent genes. The actin sensing and signaling pathway appears to operate also in fat body cells, which are proposed to produce Upd3 that then functions in an autocrine or paracrine manner to activate the Domeless-JAK-STAT pathway in the fat body. However, *Drosophila* lack an apparent DNGR1 homologue, and the sensor/receptor for actin in *Drosophila* remains unknown. The role of the actin-sensing pathway in infection or sterile inflammation also remains to be determined in part because it is not practical to perform loss of function studies of actin, which is an essential gene; identification of the actin receptor in *Drosophila* may enable such genetic studies. Nevertheless, this is an important study because it provides convincing evidence for the existence of an innate immune system that detects actin as a DAMP in *Drosophila*. This work establishes *Drosophila* as a model system to study sterile inflammation and suggests that actin is an evolutionarily conserved danger molecule from dead or damaged cells.

While the genetic dissection of these responses is impressive and compelling, there are conceptual and technical aspects of this work that need to be addressed.

Essential revisions:

1) The experiments performed demonstrate that actin-induced *TotM* expression is dependent on the JAK/STAT pathway. This finding leads to the reasonable conclusion that actin signaling proceeds via JAK/STAT. While this conclusion is certainly correct, the experiment performed was almost guaranteed to work, as the authors used a STAT-dependent gene in their analysis (*TotM*). It would be highly informative for the authors to also examine actin-inducible genes that are not annotated as STAT-response genes, and genes whose expression is repressed upon actin injection. This analysis would allow the authors to determine if JAK/STAT signaling is the only means by which actin promotes transcriptional responses. RNAseq analysis of flies deficient for STAT92E would be an ideal approach to address this question, but qPCR analysis of a few candidate genes should suffice.

2) This study provides evidence that there are differences at every level in the innate immune responses induced by actin in flies and mice. The receptors are different (DNGR-1 vs. receptor X in flies), the cellular response is different (cross presentation vs. STAT-dependent gene expression in flies), the responding cell types are different (dendritic cells vs fat body in flies) and the ligands are different (F-actin vs. G-actin). Despite these differences, the authors suggest an evolutionarily conserved role of actin as a DAMP. Can the authors really say that their evidence supports this evolutionary argument? The only evidence of evolutionary conservation is at the level of the Syk/Shark kinase and the role of ROS.

The authors could back off their argument that this study provides evidence for a conserved need for actin detection, especially in light of their lack of evidence for the importance of actin-dependent responses. Alternatively, the authors could place this data in the context of other innate immune responses that occur differently in different organisms. Although not a DAMP, LPS responses are a nice example here, as different receptors detect this PAMP in different organisms, and induce appropriate responses for that particular host (see literature of LPS responses in amoebe, mammals and plants).

3) The authors should seriously consider the possible role of ATP as the actual DAMP that they are studying. As the authors know, ATP (like actin) is abundant in healthy cells and is a well-recognized DAMP in mammalian cells. Since actin is an ATP binding protein, and the commercial source of the actin used in this study states that the actin solutions contain ATP, this is a legitimate issue. The authors are encouraged to determine if ATP injection into flies elicits any STAT-dependent gene expression, and if ATP-free solutions of actin retain activity. Along the same line, the authors should test if denatured actin (by heat and/or chemicals) or another cytoskeletal protein (e.g., tubulin) could activate the STAT pathway in flies.

4) There is no direct evidence that actin is a 'ligand' that activates the Syk-Src pathway in *Drosophila*. It is formally possible that actin injected into the flies is metabolized, modified or somehow causes tissue injury, which indirectly activates the Src pathway. Thus, some direct evidence, such as that obtained by stimulating fat body cells with actin in a cell culture setting, is necessary to establish actin as a direct ligand.

5) In tissue injury models known to activate the JAK/STAT pathway in flies, is there a detectable level of actin in the haemolymph? It is important to clarify where/when actin may be produced in the context of tissue injury or other settings. Ideally, the authors should provide evidence for the importance of actin signaling in a physiological or pathological setting.

6) The authors should provide direct evidence that Upd3 is induced by actin in *Drosophila* and hopefully in the fat body.

7) For most of the graphs in Figure 2, no error bars are indicated. Please adjust.

---

## [Author Response]

*[…] Essential revisions:*

*1) The experiments performed demonstrate that actin-induced TotM expression is dependent on the JAK/STAT pathway. This finding leads to the reasonable conclusion that actin signaling proceeds via JAK/STAT. While this conclusion is certainly correct, the experiment performed was almost guaranteed to work, as the authors used a STAT-dependent gene in their analysis (TotM). It would be highly informative for the authors to also examine actin-inducible genes that are not annotated as STAT-response genes, and genes whose expression is repressed upon actin injection. This analysis would allow the authors to determine if JAK/STAT signaling is the only means by which actin promotes transcriptional responses. RNAseq analysis of flies deficient for STAT92E would be an ideal approach to address this question, but qPCR analysis of a few candidate genes should suffice.*

We thank the reviewers for this useful suggestion. We did not mean to imply that STAT was the only transcription factor involved in the response to extracellular actin. Therefore, as the reviewers may have predicted, when we knocked down STAT in the fat body by RNAi (c564-G80^ts^>UAS-Stat IR) we observed variable levels of STAT-dependence of the genes that, according to the RNAseq data, were most up- or downregulated upon actin injection, (new Figure 4—figure supplement 1). We have therefore clarified in the revised text that the response to extracellular actin likely involves additional pathways besides fat body STAT activation.

Even STAT-dependent genes (e.g., *TotA*) rely on additional transcription factors (e.g. Relish) for their induction (Agaisse et al., 2003). This led us to carry out an additional experiment that was perhaps implicit in the reviewers’ suggestion, namely whether the (likely) contribution of additional transcription factors reflects a compound response to actin and something else. The obvious possibility is that commensal organisms or their products might be required, together with actin, to induce *TotM*. We therefore include new data in the revised paper showing that this is not the case. Indeed, the levels of TotM induced by actin are equivalent between germ-free and commensal-bearing flies. The same is true when we compared the *TotM* response between flies lacking or containing *Wolbachia*, a vertically-transmitted intracellular bacterium. These new data (new Figure 2) bolster our conclusion that we are studying a truly sterile response and that STAT activation and that of any other transcription factors contributing to the actin response are being elicited by the stimulus being tested.

*2) This study provides evidence that there are differences at every level in the innate immune responses induced by actin in flies and mice. The receptors are different (DNGR-1 vs. receptor X in flies), the cellular response is different (cross presentation vs. STAT-dependent gene expression in flies), the responding cell types are different (dendritic cells vs fat body in flies) and the ligands are different (F-actin vs. G-actin). Despite these differences, the authors suggest an evolutionarily conserved role of actin as a DAMP. Can the authors really say that their evidence supports this evolutionary argument? The only evidence of evolutionary conservation is at the level of the Syk/Shark kinase and the role of ROS.*

*The authors could back off their argument that this study provides evidence for a conserved need for actin detection, especially in light of their lack of evidence for the importance of actin-dependent responses. Alternatively, the authors could place this data in the context of other innate immune responses that occur differently in different organisms. Although not a DAMP, LPS responses are a nice example here, as different receptors detect this PAMP in different organisms, and induce appropriate responses for that particular host (see literature of LPS responses in amoebe, mammals and plants).*

We thank the reviewers for this comment. As suggested, we have changed the Introduction and Discussion to emphasise the fact that there are multiple differences between elements of the innate response to actin between flies and Man. At the same time, we have bolstered our conclusions by including a new figure to document the fact that there is something unique about the ability of actin to trigger a response in flies as we do not see a response to several other mammalian DAMPs (see new Figure 2—figure supplement 2). Finally, we now specifically refer to the suggested example of LPS, as well as toβ-glucan, to underscore the notion that inflammatory triggers can be conserved yet induce different responses in different organisms. We hope that the reviewers will agree that our manuscript now offers a more balanced narrative even if it is still written from the perspective of actin as a conserved DAMP.

*3) The authors should seriously consider the possible role of ATP as the actual DAMP that they are studying. As the authors know, ATP (like actin) is abundant in healthy cells and is a well-recognized DAMP in mammalian cells. Since actin is an ATP binding protein, and the commercial source of the actin used in this study states that the actin solutions contain ATP, this is a legitimate issue. The authors are encouraged to determine if ATP injection into flies elicits any STAT-dependent gene expression, and if ATP-free solutions of actin retain activity. Along the same line, the authors should test if denatured actin (by heat and/or chemicals) or another cytoskeletal protein (e.g., tubulin) could activate the STAT pathway in flies.*

As shown in the new Figure 2—figure supplement 2, ATP does not induce *TotM* in *Drosophila*. Furthermore, as suggested, we now show that denatured actin does not trigger the response (new Figure 2—figure supplement 2). Together, these data show that the induction of *TotM* is not due to a non-protein contaminant such as ATP in our actin preparations.

*4) There is no direct evidence that actin is a 'ligand' that activates the Syk-Src pathway in Drosophila. It is formally possible that actin injected into the flies is metabolized, modified or somehow causes tissue injury, which indirectly activates the Src pathway. Thus, some direct evidence, such as that obtained by stimulating fat body cells with actin in a cell culture setting, is necessary to establish actin as a direct ligand.*

We agree with the reviewers that an experiment with ex vivo stimulated fat body cells would be an ideal means to prove that actin engages a particular receptor. Unfortunately, we know of no available fat body cell line and we have failed to succeed in culturing fat body cells ex vivo (not shown). We attempted to stimulate multiple cell lines (e.g. S2, S1, S2R^+^, Mbn2, Kc_167_, GM2, and Cl.8) but have seen no *TotM* induction in response to actin despite validating that such lines have a competent JAK/STAT pathway as they can respond to Upd1. This suggests that these cell lines lack the response to extracellular actin, perhaps because many are haemocyte-derived and, as we show, haemocytes are dispensable for actin-induced STAT activation in vivo. The current lack of an experimental system therefore prevents us from dissecting the actin-sensing pathway in vitro. The possibility that actin is not a ligand but causes a perturbation in vivo that couples to JAK/STAT activation indirectly is one that we had already contemplated in the previous version of our manuscript and have re-iterated in the revised Discussion.

*5) In tissue injury models known to activate the JAK/STAT pathway in flies, is there a detectable level of actin in the haemolymph? It is important to clarify where/when actin may be produced in the context of tissue injury or other settings. Ideally, the authors should provide evidence for the importance of actin signaling in a physiological or pathological setting.*

We thank the reviewers for encouraging us to perform such experiments. We now show that, when flies are subjected to septic injury, actin is released into the haemolymph (New Figure 8). We further show that this induces *TotM* expression that is reduced or blocked by fat body-specific Nox or Src42A knockdown. Therefore, septic injury triggers actin release and a *TotM* response that depends on the same organ and the same molecular players that we implicated in the response to actin injection. While this is by necessity correlative (we cannot knockdown actin in vivo), it indicates that actin release does occur in vivo upon injury and that injury naturally triggers the pathway that we have uncovered using actin injection.

*6) The authors should provide direct evidence that Upd3 is induced by actin in Drosophila and hopefully in the fat body.*

The reviewers raise a point that has been vexing us for some time. We have used multiple approaches to try to measure Upd3 including qPCR using two pairs of primers (Agaisse et al., 2003; Woodcock et al., 2015) and digital droplet PCR, which is superior to conventional qPCR regarding detection and quantification of rare transcripts (Hindson et al., 2011). Neither of these approaches has yielded reproducible data. After consulting with other groups in the field, we have learned that measurement of Upd3 by qPCR is not wholly reliable. Therefore, we also attempted microscopy using a Upd3-GFP line but did not get a signal in response to actin injection. Finally, we tried Western blot using two published anti-Upd3 antibodies(Chakrabarti, Poidevin and Lemaitre, 2014; Oldefest et al., 2013). These detected a protein whose size does not match that predicted for Upd3. As none of these approaches gave conclusive results in our hands, we opted to only include the genetic evidence for a role of Upd3 in actin-induced JAK/STAT activation.

*7) For most of the graphs in Figure 2, no error bars are indicated. Please adjust.*

We thank the reviewers for pointing out this omission. We have since updated the figure in question.